# *IV-Mixed Sampler*: Leveraging Image Diffusion Models for Enhanced Video Synthesis

**Shitong Shao**[1,◇]   **Zikai Zhou**[1,◇]   **Lichen Bai**[1]   **Haoyi Xiong**[2]   **Zeke Xie**[1,*]

[1]Hong Kong University of Science and Technology (Guangzhou) [2]Baidu Inc.
`{sshao213,zikaizhou,lichenbai,zekexie}@hkust-gz.edu.cn`
`haoyi.xiong.fr@ieee.org`, ◇:Equal Contribution, *:Corresponding author

## Abstract

Exploring suitable solutions to improve performance by increasing the computational cost of inference in visual diffusion models is a highly promising direction. Sufficient prior studies have demonstrated that correctly scaling up computation in the sampling process can successfully lead to improved generation quality, enhanced image editing, and compositional generalization. While there have been rapid advancements in developing inference-heavy algorithms for improved image generation, relatively little work has explored inference scaling laws in video diffusion models (VDMs). Furthermore, existing research shows only minimal performance gains that are perceptible to the naked eye. To address this, we design a novel training-free algorithm *IV-Mixed Sampler* that leverages the strengths of image diffusion models (IDMs) to assist VDMs surpass their current capabilities. The core of *IV-Mixed Sampler* is to use IDMs to significantly enhance the quality of each video frame and VDMs ensure the temporal coherence of the video during the sampling process. Our experiments have demonstrated that *IV-Mixed Sampler* achieves state-of-the-art performance on 4 benchmarks including UCF-101-FVD, MSR-VTT-FVD, Chronomagic-Bench-150/1649, and VBench. For example, the open-source Animatediff with *IV-Mixed Sampler* reduces the UMT-FVD score from 275.2 to 228.6, closing to 223.1 from the closed-source Pika-2.0. Our code is released at https://github.com/xie-lab-ml/IV-mixed-Sampler. Our project page can be found in https://klayand.github.io/IVmixedSampler.

## 1 Introduction

In the large foundation models era, maximizing the inference potential of foundation models (Blattmann et al., 2023; Touvron et al., 2023) that require high pre-training costs has become a research staple for academics (Wu et al., 2023; Wei et al., 2022). Efficient plug-and-play algorithms (Zhang et al., 2023; Mou et al., 2024) can significantly drive large-scale models to reach their full potential and outperform the original counterparts due to low trial-and-error costs. In contrast to popular inference-heavy algorithms (Wei et al., 2022; Chen et al., 2023b; Chefer et al., 2023; Chen et al., 2024b), which consistently emerge in the large language model (LLM) and image diffusion model (IDM) field, text-to-video (T2V) synthesis still faces the prevalent challenge of low-quality synthesized videos that lack semantic faithfulness (Guo et al., 2023; Wang et al., 2023). This limitation severely hampers the deployment and application of video diffusion models (VDMs). Motivated by the success of inference scaling laws in both LLMs and IDMs, we inevitably wonder if it is possible to design an inference-heavy algorithm that is effective and plug-and-play on VDMs, allowing us to enhance VDMs' inference performance? This answer is obvious as VDM and IDM are theoretically identical (Song et al., 2023c). However, designing unique and outstanding training-free algorithms based on the properties of VDM remains a significant challenge.

**Motivation.** VDM-based training-free algorithms typically face a significant challenge: their performance ceiling is constrained by the VDM itself. To be specific, the videos synthesized from most open-source VDMs (Guo et al., 2023; Wang et al., 2023) exhibit several inherent problems, such as weak semantic consistency between the videos and the prompts, as well as low quality. A widely accepted perspective (Wu et al., 2023; Zhai et al., 2024) on this phenomenon is that underper-

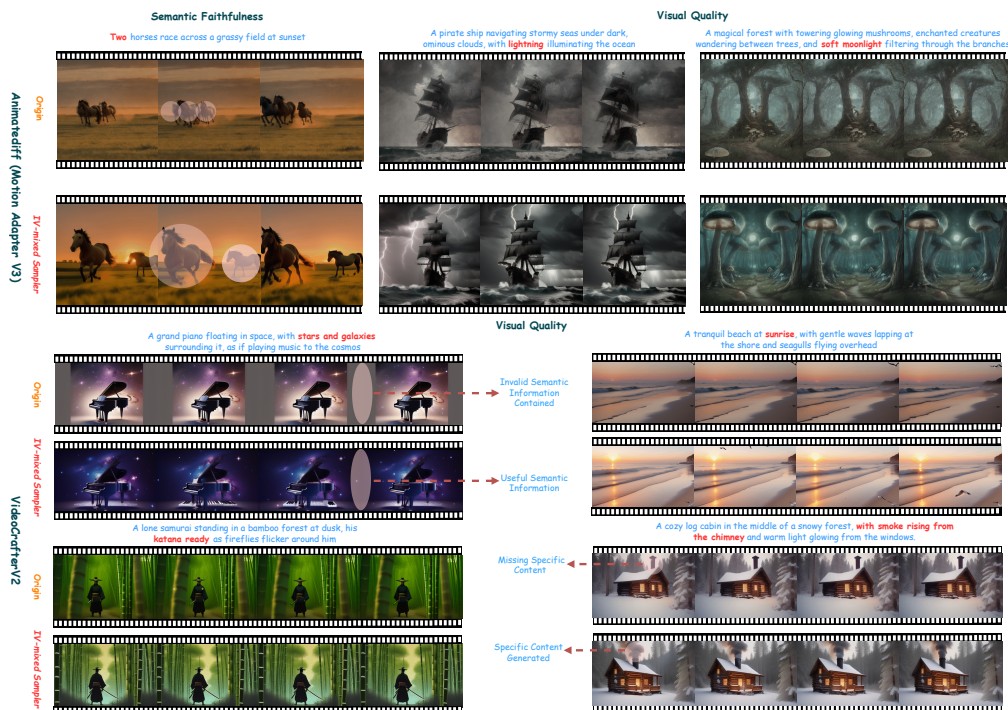

Figure 1: **Visualization of *IV-mixed Sampler* and the standard DDIM sampling on Animatediff and VideoCrafterV2.** Unlike prior heavy-inference approaches (Guo et al., 2024; Wu et al., 2023), *IV-mixed Sampler* is able to significantly improve the fidelity of the video while guaranteeing semantic faithfulness.

forming VDMs cannot overcome their intrinsic limitations because they are trained on low-quality, small-scale T2V datasets. In contrast, IDMs can now reliably serve commercial application scenarios (Betker et al., 2023; Liu et al., 2024a), thanks to the well-established dataset ecosystem created by the AIGC community. This discrepancy naturally prompts us to consider whether we can view IDMs as tools to enhance VDMs and how to make IDMs effective assistants for VDMs.

Since diffusion models employ a multi-step sampling mechanism and the quality of synthesized samples from IDMs is significantly higher than that from VDMs, it is reasonable to use IDMs and VDMs alternately for score function estimation throughout the reverse sampling process. However, this straightforward paradigm leads to a loss of temporal coherence in the synthesized videos during our initial empirical explorations. Inspired by SDEdit (Meng et al., 2021), we enhance the quality of each frame at every denoising step by performing the following additional operation: *1) first adding Gaussian noise and 2) then denoising using IDMs.* We found that this form of inference significantly impairs performance. Through empirical exploration, we identified the issue as the "first adding Gaussian noise" step, which introduces excessive randomness. Replacing this step with a deterministic modeling paradigm (*i.e.*, *deterministic sampling*) yields stable performance gains.

In this paper, we consider the widely used *deterministic function* DDIM-Inversion (Mokady et al., 2023) to inject perturbations into the video. To be specific, we integrate IDM and VDM using the base operator: first performing DDIM-Inversion, followed by applying operations similar to DDIM before each denoising step. Additionally, the observation that performing DDIM-Inversion with IDM first significantly outperforms performing DDIM-Inversion with VDM first further supports the conclusion that IDMs trained on high-quality datasets can positively impact the video sampling process. Motivated by this, we further explore the upper bound of the performance gain that IDM provides for video synthesis. We primarily extend the paradigm of the single-step diffusion process and the single-step reverse process to multiple steps. Then, to ensure inference efficiency, we investigate all possible combinations in the two-step diffusion process and the two-step inverse process, collectively naming this series of algorithms *IV-mixed Sampler*.

**Contribution.** Specifically, *1)* we construct *IV-mixed Sampler* under a rigorous mathematical framework and demonstrate, through theoretical analysis, that it can be elegantly transformed into a standard inverse ordinary differential equation (ODE) process. For the sake of intuition, we

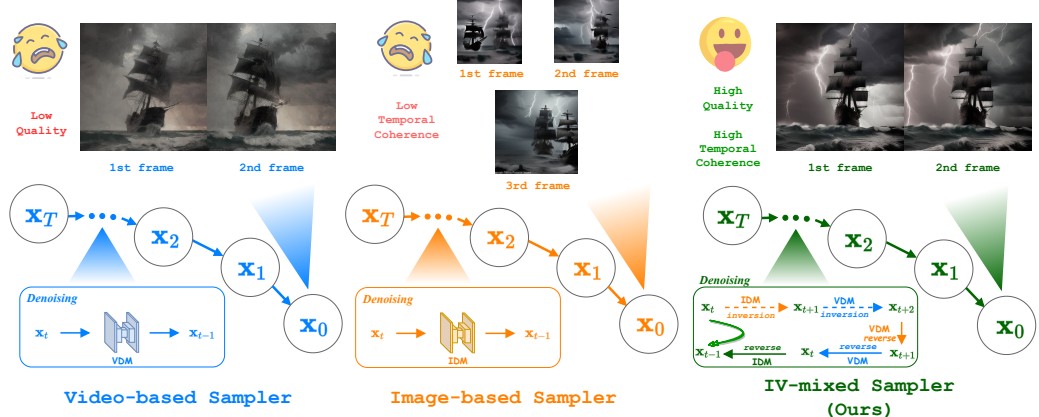

Figure 2: **Overview of our *IV-mixed Sampler*, Video-based Sampler and Image-based Sampler.** *IV-mixed Sampler* utilizes IDM and VDM to ensure synthesized video quality and temporal coherence, respectively.

present *IV-mixed Sampler* (*w.r.t.*, "IV-IV") on Fig. 2 and its pseudo code in Appendix B. *2)* The empirically optimal *IV-mixed Sampler* further reduces the UMT-FVD by approximately by 39.72 points over FreeInit (Wu et al., 2023). Furthermore, *3)* we conduct sufficient ablation studies to determine which classifier-free guidance (CFG) (Ho & Salimans, 2021) scale and which sampling paradigm yield the best performance for various metrics at what sampling intervals. In addition to this, *4)* qualitative and quantitative comparison experiments have amply demonstrated that our algorithm achieves state-of-the-art (SOTA) performance on five popular benchmarks: UCF-101-FVD (Soomro, 2012), MSR-VTT-FVD (Xu et al., 2016), Chronomagic-Bench-150 (Yuan et al., 2024), Chronomagic-1649 (Yuan et al., 2024) and VBench (Huang et al., 2024). These competitive outcomes demonstrate that our proposed *IV-mixed Sampler* dramatically improves the visual quality and semantic faithfulness of the synthesized video.

## 2 PRELIMINARY

We review VDMs, SDEdit, DDIM & DDIM-Inversion in this section and further bootstrap how past work has designed VDM-based plug-and-play algorithms in Appendix A.3.

**Diffusion Models.** Diffusion models (Ho et al., 2020; Song et al., 2023c;a) including IDMs and VDMs consist of a forward process and a reverse process. Given $\mathbf{x}_0$ represent a $D$-dimensional random variable sampled from the real data distribution $q_0(\mathbf{x}_0)$. The forward process injects Guassian noise $\epsilon_t \sim \mathcal{N}(0, \mathbf{I})$ to the clean data as follows:

$$\mathbf{x}_t = \alpha_t \mathbf{x}_0 + \sigma_t \epsilon_t, \tag{1}$$

where $t \sim \mathcal{U}[\eta, 1]$ ($\eta$ is a very small quantity defaulting to 1e-5) and $\alpha_t$ and $\sigma_t$ are components of a predefined noise schedule. $\sigma_t$ is monotonically increasing from 0 to 1 in all diffusion models, while $\alpha_t$ can either remain constant (*e.g.*, EDM (Karras et al., 2022)) or decrease monotonically (*e.g.*, VP-SDE (Song et al., 2023c) and Rectified Flow (Liu et al., 2022)) from 0 to 1. The forward process in Eq. 1 can be rewritten as the following stochastic differential equation (SDE):

$$d\mathbf{x}_t = f(t)\mathbf{x}_t dt + g(t)\overline{\boldsymbol{\omega}}_t, \tag{2}$$

where $f(t)$ and $g(t)$ denote the *drift coefficient* of $\mathbf{x}_t$ and the *diffusion coefficient* of $\mathbf{x}_t$, respectively. $\overline{\boldsymbol{\omega}}_t$ refers to a standard Wiener process. Eq. 2 has a corresponding ODE-based reverse process defined as:

$$d\mathbf{x}_t = f(t)\mathbf{x}_t - \frac{1}{2}g^2(t)\nabla_{\mathbf{x}} \log q_t(\mathbf{x}), \tag{3}$$

where $\nabla_{\mathbf{x}} \log q_t(\mathbf{x})$ denotes the score function $\nabla_{\mathbf{x}_t} \log p_t(\mathbf{x}_t)$. Since $\nabla_{\mathbf{x}} \log q_t(\mathbf{x})$ cannot be accessed during the reverse process, it must be replaced by a linear transformation $\frac{\epsilon_\theta(\mathbf{x}_t, t)}{-\sigma_t}$ using the noise estimation model $\epsilon_\theta(\cdot, \cdot)$ or the score function estimation model $\mathbf{s}_\theta(\cdot, \cdot)$.

**Video Diffusion Model vs. Image Diffusion Model.** The most critical differences between IDM and VDM lie in their training sets and model architectures. Regarding training sets, IDM typically utilizes large-scale, high-quality datasets, enabled by the increasing availability of both real and synthetic image datasets (Schuhmann et al., 2022). Conversely, the performance of VDMs is significantly constrained by the limited number of publicly available video datasets, such as Webvid-10M (Bain et al., 2021) and Pandas-70M (Chen et al., 2024c), which contain low-quality real data. In terms of model architectures, several VDMs differ from their IDM counterparts because video data $\mathbf{x}_0 \in \mathbb{R}^{b \times c \times t \times h \times w}$ includes an additional time dimension t, unlike image data $\mathbf{x}_0 \in \mathbb{R}^{b \times c \times h \times w}$, where b, c, h, and w refer to batch size, number of channels, height, and width, respectively. Therefore, most VDMs incorporate both spatial and temporal blocks, where the spatial block aligns with the module used in the mainstream IDMs, while the temporal block employs 3D convolutions (Guo et al., 2023) or tokens in the time dimension (Ma et al., 2024) to capture temporal information.

**SDEdit.** SDEdit (Meng et al., 2021) is an image editing method that produces a latent variable capable of reconstructing the input $\mathbf{x}_0$. It first injects Gaussian noise into $\mathbf{x}_0$, then performs editing in the latent space, and finally obtains pure samples through a standard reverse process. This approach provides a scheme for editing the latent variable using IDM at each step of sampling. Since SDEdit is nontrivial to invert, we employ DDIM-Inversion (Mokady et al., 2023) to map the latent $\mathbf{x}_t$ back to its previous counterpart $\mathbf{x}_{t+\Delta t}$, where $1 - t \geq \Delta t > 0$.

**DDIM & DDIM-Inversion.** DDIM (Song et al., 2023a) serves as an efficient ODE-based sampler that iteratively refines the initial Gaussian noise $\mathbf{x}_T$, ultimately generating a "clean" video $\mathbf{x}_0$ that conforms to the data distribution $p_0(\mathbf{x}_0)$. A notable characteristic of DDIM is its adherence to *deterministic sampling*. The synthesized data produced by DDIM can be expressed as follows:

$$\mathbf{x}_t = \mathcal{L}(\mathbf{x}_s) = \alpha_t \left( \frac{\mathbf{x}_s - \sigma_s \boldsymbol{\epsilon}_\theta(\mathbf{x}_s, s)}{\alpha_s} \right) + \sigma_t \boldsymbol{\epsilon}_\theta(\mathbf{x}_s, s). \tag{4}$$

In this equation, $s$ and $t$ denote timesteps $\in \mathcal{U}[0, 1]$, with the condition that $t \leq s$. When under the constraint $t \geq s$, the standard DDIM in Eq. 4 is referred to as DDIM-Inversion $\mathbf{x}_t = \mathcal{L}^{-1}(\mathbf{x}_s)$. In practical, DDIM is often incorporated into classifier-free guidance for more precise control as well as higher quality data synthesis. Thus, Eq. 4 can be transferred into

$$\mathbf{x}_t = \mathcal{L}(\mathbf{x}_s, \omega) = \left[ \alpha_t \mathbf{x}_s + (\alpha_s \sigma_t - \alpha_t \sigma_s)[(\omega + 1)\boldsymbol{\epsilon}_\theta(\mathbf{x}_s, s, \mathbf{c}) - \omega \boldsymbol{\epsilon}_\theta(\mathbf{x}_s, s, \varnothing)]] / \alpha_s, \tag{5}$$

where $\mathbf{c}$, $\varnothing$ and $\omega$ stand for the text prompt, the null prompt and the CFG scale, respectively.

Additionally, our proposed *IV-mixed Sampler*, along with Z-Sampling (LiChen et al., 2025) and Golden Noise (Zhou et al., 2024) applied in the image domain, uses DDIM and DDIM-Inversion to inject semantic information, which is essentially a noise optimization framework (Qi et al., 2024). Unlike these approaches, which consider how to perform more efficient resampling operations, *IV-mixed Sampler* focuses on utilizing both IDM and VDM to maximize the visual quality and motion consistency of the synthesized video.

## 3 APPROACH

Observing that IDMs produce high-quality samples without ensuring temporal coherence, while VDMs ensure temporal continuity but generate low-quality video, we propose *IV-mixed Sampler* to combine the strengths of both IDMs and VDMs (see Fig. 2). In this section, we first describe how to sample from $\mathbf{x}_t$ to $\mathbf{x}_{t+\Delta t}$ and from $\mathbf{x}_{t+\Delta t}$ to $\mathbf{x}_t$ using IDM and VDM. We then introduce *IV-mixed Sampler* and outline its design space of hyperparameters, followed by a theoretical analysis. Finally, we discuss the effect of sampling in the latent space on *IV-mixed Sampler*.

### 3.1 FORWARD (GO!!) AND REVERSE (BACK!!)

A crucial step in overcoming the bottleneck of VDM $\epsilon_\theta^V(\cdot, \cdot)$ with IDM $\epsilon_\theta^I(\cdot, \cdot)$ is to address the domain gap between IDM and VDM caused by differences in training data. Therefore, a specialized rescheduling paradigm is required to achieve the mapping $\mathbf{x}_t \to \mathcal{L}^{-1}(\mathbf{x}_t) \to \mathcal{L}(\mathcal{L}^{-1}(\mathbf{x}_t)) = \mathbf{x}_t$, where $\mathcal{L}(\cdot)$ is a *deterministic* function with an inverse $\mathcal{L}^{-1}(\cdot)$. Given this, we can modify $\mathbf{x}_{x+\Delta t} \approx$

Table 1: **Quantitative comparison with popular heavy-inference algorithms on Chronomagic-Bench-150.** $(+\Delta\,[\cdot])$ stands for the improvement of *IV-mixed Sampler* compared to the "Standard DDIM". I4VGen is not compared across other architectures because its official implementation is limited to Animatediff.

| Model | Method | Extra Model | UMT-FVD ($\downarrow$) | UMTScore ($\uparrow$) | GPT4o-MTScore ($\uparrow$) |
|---|---|---|---|---|---|
| VideoCrafterV2 | Standard DPM-Solver++ | ✗ | $214.06_{\pm0.007}$ | $2.77_{\pm0.014}$ | $2.85_{\pm0.025}$ |
| | *IV-mixed Sampler* (ours) | ✓ | $210.58_{\pm0.010}$ $(+\Delta\,3.48)$ | $3.29_{\pm0.011}$ $(+\Delta\,0.51)$ | $3.30_{\pm0.006}$ $(+\Delta\,0.45)$ |
| ModelScope-T2V | Standard DDIM | ✗ | $241.61_{\pm0.002}$ | $2.65_{\pm0.017}$ | $2.97_{\pm0.013}$ |
| | FreeInit | ✗ | $220.96_{\pm0.002}$ | $2.99_{\pm0.025}$ | $3.10_{\pm0.015}$ |
| | *IV-mixed Sampler* (ours) | ✓ | $234.90_{\pm0.001}$ $(+\Delta\,6.71)$ | $3.00_{\pm0.022}$ $(+\Delta\,0.35)$ | $3.16_{\pm0.035}$ $(+\Delta\,0.19)$ |
| Animatediff (SD V1.5, Motion Adapter V3) | Standard DDIM | ✗ | $275.18_{\pm0.008}$ | $2.81_{\pm0.030}$ | $2.86_{\pm0.026}$ |
| | FreeInit | ✗ | $268.31_{\pm0.005}$ | $2.85_{\pm0.022}$ | $2.60_{\pm0.026}$ |
| | I4VGen | ✓ | $227.22_{\pm0.010}$ | $2.68_{\pm0.030}$ | $3.01_{\pm0.017}$ |
| | *IV-mixed Sampler* (ours) | ✓ | $228.60_{\pm0.008}$ $(+\Delta\,46.58)$ | $3.32_{\pm0.018}$ $(+\Delta\,0.51)$ | $3.55_{\pm0.028}$ $(+\Delta\,0.69)$ |

Table 2: **Quantitative comparison with popular heavy-inference algorithms, including FreeInit and I4VGen, on Chronomagic-Bench-1649 (Yuan et al., 2024).** Yellow cells stands for the winner.

| Model | Method | Extra Model | UMT-FVD ($\downarrow$) | UMTScore ($\uparrow$) | GPT4o-MTScore ($\uparrow$) |
|---|---|---|---|---|---|
| VideoCrafterV2 | Standard DPM-Solver++ | ✗ | $178.44_{\pm0.000}$ | $2.75_{\pm0.001}$ | $2.69_{\pm0.021}$ |
| | *IV-mixed Sampler* (ours) | ✓ | $172.03_{\pm0.004}$ $(+\Delta\,6.41)$ | $3.35_{\pm0.003}$ $(+\Delta\,0.60)$ | $3.05_{\pm0.028}$ $(+\Delta\,0.36)$ |
| ModelScope-T2V | Standard DDIM | ✗ | $199.52_{\pm0.001}$ | $2.99_{\pm0.004}$ | $3.16_{\pm0.026}$ |
| | *IV-mixed Sampler* (ours) | ✓ | $197.44_{\pm0.000}$ $(+\Delta\,2.08)$ | $3.06_{\pm0.003}$ $(+\Delta\,0.07)$ | $3.24_{\pm0.023}$ $(+\Delta\,0.08)$ |
| Animatediff (SD V1.5, Motion Adapter V3) | Standard DDIM | ✗ | $219.29_{\pm0.001}$ | $3.08_{\pm0.004}$ | $2.62_{\pm0.016}$ |
| | FreeInit | ✗ | $209.60_{\pm0.004}$ | $3.08_{\pm0.004}$ | $2.72_{\pm0.014}$ |
| | I4VGen | ✓ | $206.22_{\pm0.003}$ | $3.21_{\pm0.009}$ | $3.09_{\pm0.031}$ |
| | *IV-mixed Sampler* (ours) | ✓ | $192.72_{\pm0.001}$ $(+\Delta\,26.57)$ | $3.39_{\pm0.007}$ $(+\Delta\,0.31)$ | $3.35_{\pm0.022}$ $(+\Delta\,0.73)$ |

$\mathcal{L}^{-1}(\mathbf{x}_t)$ without disrupting the standard sampling process. We utilize DDIM-Inversion and DDIM to implement $\mathcal{L}^{-1}(\cdot,\cdot)$ and $\mathcal{L}(\cdot,\cdot)$, respectively. By introducing CFG, we can define a new paradigm to implement IDMs' information injection via the operator G, *i.e.* semantic information injection:

$$\mathbf{x}_t' = \mathcal{I}(\mathbf{x}_t, \omega_{\text{go}}, \omega_{\text{back}}, \mathtt{G}, \epsilon_\theta^{\text{go}}, \epsilon_\theta^{\text{back}}) = \mathcal{L}(\mathbf{u} + \mathtt{G}(\mathbf{u}), \omega_{\text{back}}), \text{ where } \mathbf{u} = \mathcal{L}^{-1}(\mathbf{x}_t, \omega_{\text{go}}). \quad (6)$$

In Eq. 6, $\omega_{\text{go}}$ and $\omega_{\text{back}}$ stand for the CFG scales for the DDIM-Inversion and DDIM, respectively. $\epsilon_\theta^{\text{go}}(\cdot,\cdot)$ and $\epsilon_\theta^{\text{back}}(\cdot,\cdot)$ are the noise estimation models used to perform $\mathcal{L}^{-1}(\cdot,\cdot)$ and $\mathcal{L}(\cdot,\cdot)$, respectively. G is any function whose mission is to make modifications to $\mathbf{u}$. It is worth noting that for IDMs, we first reshape $\mathbf{x}_t$ from the shape $b\times c\times t\times h\times w$ to $(b\times t)\times c\times h\times w$, and then pass it through the noise estimation model to perform the operation $\mathcal{I}$ in the practical implementation. If $\omega_{\text{go}} \equiv \omega_{\text{back}}$, $\epsilon_\theta^{\text{go}}(\cdot,\cdot) \equiv \epsilon_\theta^{\text{back}}(\cdot,\cdot)$ and $\mathtt{G}(\cdot)$ is an identity operator, then $\mathbf{x}_t' \equiv \mathbf{x}_t$. To understand how $\mathcal{I}$ injects semantic information, we rewrite Eq. 7 using a first-order Taylor expansion as

$$\mathcal{I}(\mathbf{x}_t, \omega_{\text{go}}, \omega_{\text{back}}, \mathtt{G}, \epsilon_\theta^{\text{go}}, \epsilon_\theta^{\text{back}}) = \mathcal{L}(\mathcal{L}^{-1}(\mathbf{x}_t, \omega_{\text{go}}) + \mathtt{G}(\mathcal{L}^{-1}(\mathbf{x}_t, \omega_{\text{go}})), \omega_{\text{back}}) = \mathcal{L}(\mathcal{L}^{-1}(\mathbf{x}_t, \omega_{\text{go}}), \omega_{\text{go}})$$

$$+ (\omega_{\text{back}} - \omega_{\text{go}})\frac{\partial\mathcal{L}(\mathcal{L}^{-1}(\mathbf{x}_t, \omega_{\text{go}}), \omega_{\text{go}})}{\partial\omega_{\text{go}}} + \mathtt{G}(\mathcal{L}^{-1}(\mathbf{x}_t, \omega_{\text{go}}))\frac{\partial\mathcal{L}(\mathcal{L}^{-1}(\mathbf{x}_t, \omega_{\text{go}}), \omega_{\text{go}})}{\partial\mathcal{L}^{-1}(\mathbf{x}_t, \omega_{\text{go}})} + \mathcal{O}((\omega_{\text{back}} - \omega_{\text{go}})^2)$$

$$+ \mathcal{O}(\mathtt{G}(\mathcal{L}^{-1}(\mathbf{x}_t, \omega_{\text{go}}))^2) = \mathbf{x}_t \quad \text{\# define } J = \alpha_{t+\Delta t}\sigma_t - \alpha_t\sigma_{t+\Delta t}$$

$$+ (\omega_{\text{back}} - \omega_{\text{go}})\frac{\partial\left(\mathbf{x}_t + [J[(\omega_{\text{back}} + 1)\epsilon_\theta^{\text{back}}(\mathbf{x}_{t+\Delta t}, t + \Delta t, \mathbf{c}) - \omega_{\text{back}}\epsilon_\theta^{\text{back}}(\mathbf{x}_{t+\Delta t}, t + \Delta t, \varnothing)]/\alpha_{t+\Delta t}\right)}{\partial\omega_{\text{go}}} \quad (7)$$

$$+ (\omega_{\text{back}} - \omega_{\text{go}})\frac{\partial\left((\alpha_t\sigma_{t+\Delta t} - \alpha_{t+\Delta t}\sigma_t)[(\omega_{\text{go}} + 1)\epsilon_\theta^{\text{go}}(\mathbf{x}_t, t, \mathbf{c}) - \omega_{\text{go}}\epsilon_\theta^{\text{go}}(\mathbf{x}_t, t, \varnothing)]/\alpha_{t+\Delta t}\right)}{\partial\omega_{\text{go}}} +$$

$$\mathtt{G}(\mathcal{L}^{-1}(\mathbf{x}_t, \omega_{\text{go}}))\frac{\partial\mathcal{L}(\mathcal{L}^{-1}(\mathbf{x}_t, \omega_{\text{go}}), \omega_{\text{go}})}{\partial\mathcal{L}^{-1}(\mathbf{x}_t, \omega_{\text{go}})} = \mathbf{x}_t + (\omega_{\text{back}} - \omega_{\text{go}})\frac{(\alpha_t\sigma_{t+\Delta t} - \alpha_{t+\Delta t}\sigma_t)[\epsilon_\theta^{\text{go}}(\mathbf{x}_t, t, \mathbf{c}) - \epsilon_\theta^{\text{go}}(\mathbf{x}_t, t, \varnothing)]}{\alpha_{t+\Delta t}}$$

$$+ \mathtt{G}(\mathcal{L}^{-1}(\mathbf{x}_t, \omega_{\text{go}}))\frac{\partial\mathcal{L}(\mathcal{L}^{-1}(\mathbf{x}_t, \omega_{\text{go}}), \omega_{\text{go}})}{\partial\mathcal{L}^{-1}(\mathbf{x}_t, \omega_{\text{go}})}. \quad \text{\# Ignore second-order and higher terms.}$$

Observing the pink term in Eq. 7, we can inject semantic information from both $\epsilon_\theta^{\text{go}}(\cdot,\cdot)$ and $\epsilon_\theta^{\text{back}}(\cdot,\cdot)$ into $\mathbf{x}_t$ by setting $\omega_{\text{back}} - \omega_{\text{go}} > 0$. In particular, we set $\omega_{\text{back}} = -\omega_{\text{go}}$ and $\omega_{\text{back}} > 0$ by default. Given this, we only need to replace $\epsilon_\theta^{\text{go}}$ with $\epsilon_\theta^{\text{I}}(\cdot,\cdot)$ or $\epsilon_\theta^{\text{V}}(\cdot,\cdot)$ to inject specific semantic information, thereby enhancing the visual quality or temporal coherence of the synthesized video.

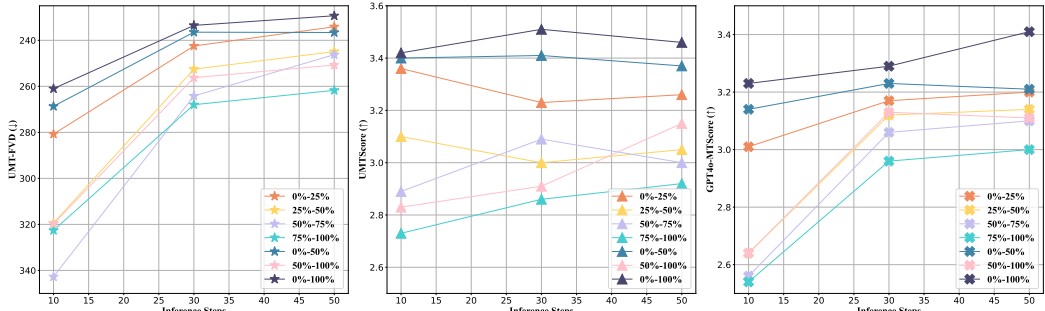

Figure 3: **Ablation studies on sampling intervals of *IV-mixed Sampler* ("IV-IV") with Animatediff (SD V1.5, Motion Adapter V3).** The BEGIN%-END% in the legend indicates the portion of the entire sampling process performed by *IV-mixed Sampler*. For example, in a 50-step sampling scenario, 0%-50% corresponds to *IV-mixed Sampler* being applied during steps 1-25. More details of "IV-VI" can be found in Appendix D.

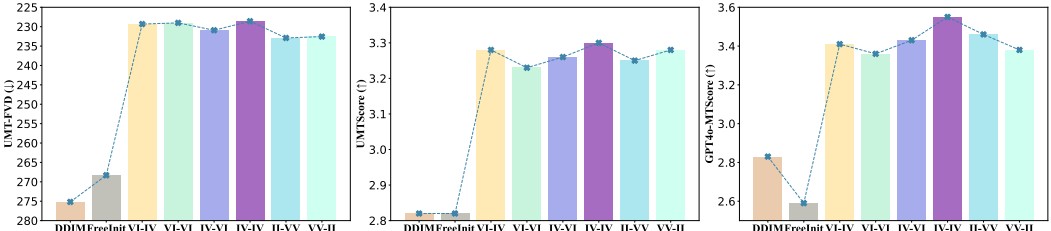

Figure 4: **Ablation studies on different $C_4^2$ species combinations with Animatediff (SD V1.5, Motion Adapter V3).** We can clearly observe that *IV-mixed Sampler* ("IV-IV") is the winner across all metrics.

## 3.2 IV-MIXED SAMPLER

The paradigm defined in Eq. 6 enables us to easily describe all the base Operators in IV-mixed Sampler. We define "R-", "I-", and "V-" as follows: *1)* "R-": Forward noise addition using random noise; *2)* "I-": Forward noise addition using DDIM Inversion with IDM; *3)* "V-": Forward noise addition using DDIM Inversion with VDM. Moreover, the front of the horizontal line "-" refers to the additive noise form, while the back of "-" represents the denoising paradigm. For instance, "I-I" and "V-V" can represent $\mathcal{I}(\mathbf{x}_t, -h, h, \mathtt{G}, \boldsymbol{\epsilon}_\theta^\mathrm{I}, \boldsymbol{\epsilon}_\theta^\mathrm{I})$ and $\mathcal{I}(\mathbf{x}_t, -h, h, \mathtt{G}, \boldsymbol{\epsilon}_\theta^\mathrm{V}, \boldsymbol{\epsilon}_\theta^\mathrm{V})$, respectively, under the condition $h > 0$ and $\mathtt{G}$ is an identity operator. Clearly, this represents only a single-step injection of semantic information. We can construct *IV-mixed Sampler* to allow multi-step injections of semantic information using $\mathtt{G}$ and the recursive definition:

$$
\begin{aligned}
\mathbf{x}_t' = \mathtt{G}^1(\mathbf{x}_t) &= \mathcal{I}(\mathbf{x}_t, -h, h, \mathtt{G}^2, \boldsymbol{\epsilon}_\theta^{1,\mathrm{go}}, \boldsymbol{\epsilon}_\theta^{1,\mathrm{back}}), \\
\mathtt{G}^2(\mathbf{y}) &= \mathcal{I}(\mathbf{y}, -h, h, \mathtt{G}^3, \boldsymbol{\epsilon}_\theta^{2,\mathrm{go}}, \boldsymbol{\epsilon}_\theta^{2,\mathrm{back}}), \\
&\vdots \\
\mathtt{G}^N(\mathbf{y}) &= \mathcal{I}(\mathbf{y}, -h, h, \mathtt{G}^{N+1}, \boldsymbol{\epsilon}_\theta^{N,\mathrm{go}}, \boldsymbol{\epsilon}_\theta^{N,\mathrm{back}}), \quad \text{s.t.} \quad N \geq 1
\end{aligned}
\tag{8}
$$

where $N$ and $\mathtt{G}^{N+1}$ refer to the number of semantic information injection and the identity operator, respectively. Through Eq. 8, we can easily represent *IV-mixed Sampler* with different $N$. For instance, "IV-VI" can be described as $\mathbf{x}_t' = \mathtt{G}^1(\mathbf{x}_t) = \mathcal{I}(\mathbf{x}_t, -h, h, \mathtt{G}^2, \boldsymbol{\epsilon}_\theta^\mathrm{I}, \boldsymbol{\epsilon}_\theta^\mathrm{I})$, $\mathtt{G}^2(\mathbf{y}) = \mathcal{I}(\mathbf{y}, -h, h, \mathtt{G}^3, \boldsymbol{\epsilon}_\theta^\mathrm{V}, \boldsymbol{\epsilon}_\theta^\mathrm{V})$ and $\mathtt{G}^3$ is an identity operator. Considering the computational overhead and performance trade-offs, this paper focuses only on the scenario where $N = 2$. In Sec. 4's ablation study, we further restrict $\boldsymbol{\epsilon}_\theta^{1,\mathrm{go}}$, $\boldsymbol{\epsilon}_\theta^{1,\mathrm{back}}$, $\boldsymbol{\epsilon}_\theta^{2,\mathrm{go}}$, and $\boldsymbol{\epsilon}_\theta^{2,\mathrm{back}}$ to two models each, occupied by $\boldsymbol{\epsilon}_\theta^\mathrm{I}(\cdot, \cdot)$ and $\boldsymbol{\epsilon}_\theta^\mathrm{V}(\cdot, \cdot)$ (*w.r.t.*, $C_2^4$ combinations), and find that "IV-IV" performs best. The visualization in Fig. 1 demonstrates that "IV-IV" significantly improves both the visual quality of the synthesized video and the consistency between the video and the text prompt.

## 3.3 DISCUSSION

**Hyperparameter Design Space.** In this paper, we elucidate three design choices: the $C_2^4$ combinations mentioned in Sec. 3.2, the intervals at which *IV-mixed Sampler* is performed during standard

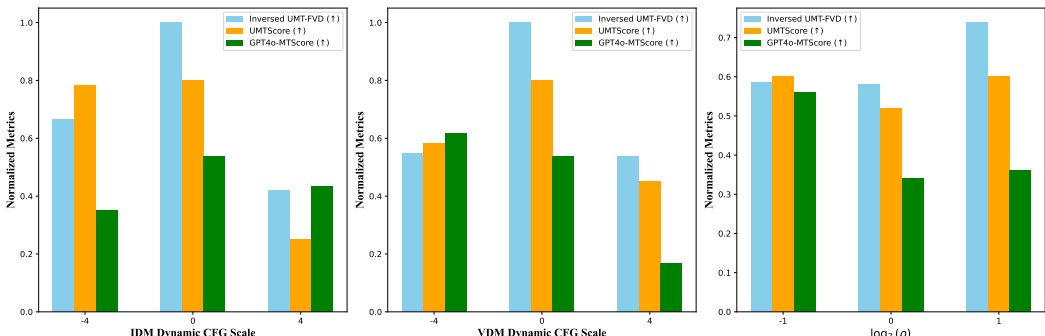

Figure 5: **Ablation studies on dynamic CFG scale with Animatediff (SD V1.5, Motion Adapter V3).** We ensure that $h^{\text{go}}(t) \equiv h^{\text{back}}(t)$. Consequently, the IDM dynamic CFG scale corresponds to IDM's $\gamma_{\text{go}}^{\text{t=0}} - \gamma_{\text{go}}^{\text{t=1}}$. Similarly, the VDM dynamic CFG scale corresponds to VDM's $\gamma_{\text{go}}^{\text{t=0}} - \gamma_{\text{go}}^{\text{t=1}}$. $\log_7(\rho)$ is 1, 0, and -1 indicates whether the change from $\gamma_{\text{go}}^{\text{t=0}}$ to $\gamma_{\text{go}}^{\text{t=1}}$ is convex, linear, or concave, respectively. All metrics are normalized. Note that we back-normalized UMT-FVD for consistency.

DDIM sampling, and the dynamic CFG scale. All three forms are expected to find the empirically optimal solution, with the first two being natural explorations and the last addressing the domain gap between IDMs and VDMs, which is expected to be mitigated by adjusting the CFG scale. For the last one, to be specific, we re-express $-h$ and $h$ in Eq. 8 as $-h^{\text{go}}(t)$ and $h^{\text{back}}(t)$. Inspired by the Karras's noise schedule (Karras et al., 2022), we define $h^{\text{go}}(t)$ and $h^{\text{back}}(t)$ as

$$h^{\text{go}}(t) = (\left[\gamma_{\text{go}}^{\text{t=0}}\right]^{\frac{1}{\rho}} + t(\left[\gamma_{\text{go}}^{\text{t=1}}\right]^{\frac{1}{\rho}} - \left[\gamma_{\text{go}}^{\text{t=0}}\right]^{\frac{1}{\rho}}))^{\rho}, h^{\text{back}}(t) = (\left[\gamma_{\text{back}}^{\text{t=0}}\right]^{\frac{1}{\rho}} + t(\left[\gamma_{\text{back}}^{\text{t=1}}\right]^{\frac{1}{\rho}} - \left[\gamma_{\text{back}}^{\text{t=0}}\right]^{\frac{1}{\rho}}))^{\rho}, \quad (9)$$

where $\gamma_{\text{go}}^{\text{t=1}}$ and $\gamma_{\text{back}}^{\text{t=1}}$ represent the CFG scales when $t = 1$, while $\gamma_{\text{go}}^{\text{t=0}}$ and $\gamma_{\text{back}}^{\text{t=0}}$ denote the CFG scales when $t = 0$. The parameter $\rho$ controls the concave and convex properties of the CFG scale curve with respect to $t$. The experiments in Sec. 4 demonstrate that the dynamic CFG scale can be adjusted to achieve performance improvements for specific aspects (*e.g.*, semantic faithfulness).

**Theoretical Analysis.** As shown in Theorem 3.1, *IV-mixed Sampler* can be elegantly transformed into an ODE, taking the same form as Eq. 3. Consequently, *IV-mixed Sampler* preserves the standard sampling process (*e.g.*, DDIM or Euler–Maruyama), enabling a trade-off between temporal coherence and visual quality by adjusting the parameters $\omega_{\text{go-back}}^{\text{IDM}}$, $\omega_{\text{go-back}}^{\text{VDM}}$, and $\omega$.

**Theorem 3.1.** *(the proof in Appendix C) IV-mixed Sampler can be transferred to an ODE. For example, the ODE corresponding to "IV-IV" is*

$$d\mathbf{x}_t = f(t)\mathbf{x}_t - \frac{1}{2}g^2(t) \left[ \omega_{\text{go-back}}^{IDM} \frac{g^2(t)}{2} \nabla_{\mathbf{x}} \log q_t^{IDM}(\mathbf{c}|\mathbf{x}) + (\omega_{\text{go-back}}^{VDM} + \omega) \frac{g^2(t)}{2} \nabla_{\mathbf{x}} \log q_t^{VDM}(\mathbf{c}|\mathbf{x}) \right], \quad (10)$$

*Here, $\omega$ refers to the vanilla CFG scale, while both $\omega_{\text{go-back}}^{IDM}$ and $\omega_{\text{go-back}}^{IDM}$ are CFG scales that are greater than 0. Let $\nabla_{\mathbf{x}} \log q_t^{IDM}(\mathbf{c}|\mathbf{x})$ and $\nabla_{\mathbf{x}} \log q_t^{VDM}(\mathbf{c}|\mathbf{x})$ represent the score function estimated for $p_t(\mathbf{x}_t)$ using IDM and VDM under classifier-free guidance.*

**The Influence of Latent Space.** Most current high-resolution IDMs and VDMs follow the latent diffusion model (LDM) paradigm (Rombach et al., 2022a). As a result, leveraging *IV-mixed Sampler* requires IDM and VDM to share the same latent space, meaning they must use the same VAE (Kusner et al., 2017). Fortunately, for most VDMs, we always can find the corresponding IDM that share the same latent space with the VDM. For cases where this condition is not met, we can convert the video from shape b×c×t×h×w to (b×t)×c×1×h×w and then pass it through VDM, which is referred to as using "IDM". This paradigm is also applied in I4VGen (Guo et al., 2024).

## 4 EXPERIMENT

In this section, we present experiments to demonstrate the effectiveness of *IV-mixed Sampler*. Specifically, we perform qualitative and quantitative comparisons across various benchmarks on multiple T2V diffusion models, including ModelScope-T2V (Wang et al., 2023), Animatediff (Guo

Table 3: **Quantitative comparison on UCF-101 datasets using Animatediff and ModelScope-T2V.** ($+\Delta$ [·]) indicates the improvement of *IV-mixed Sampler* compared to the "Standard DDIM".

| Model | Method | Extra Model | FVD ($\downarrow$) (StyleGAN) | FVD ($\downarrow$) (VideoGPT) |
|---|---|---|---|---|
| Animatediff (SD V1.5, Motion Adapter V3) | Standard DDIM | ✗ | $815.07_{\pm 0.006}$ | $819.93_{\pm 0.013}$ |
| | FreeInit | ✗ | $805.33_{\pm 0.011}$ | $807.02_{\pm 0.031}$ |
| | Unictrl | ✗ | $1859.11_{\pm 0.030}$ | $1863.32_{\pm 0.024}$ |
| | I4VGen | ✓ | $803.26_{\pm 0.024}$ | $805.75_{\pm 0.016}$ |
| | *IV-mixed Sampler* (ours) | ✓ | $800.47_{\pm 0.032}$ ($+\Delta$ 14.60) | $804.67_{\pm 0.028}$ ($+\Delta$ 15.26) |
| ModelScope-T2V | Standard DDIM | ✗ | $1492.16_{\pm 0.022}$ | $1484.98_{\pm 0.006}$ |
| | *IV-mixed Sampler* (ours) | ✓ | $841.65_{\pm 0.032}$ ($+\Delta$ 650.51) | $838.05_{\pm 0.011}$ ($+\Delta$ 656.93) |

Table 4: **Quantitative comparison on MSR-VTT using Animatediff and ModelScope-T2V.** ($+\Delta$ [·]) indicates the improvement of *IV-mixed Sampler* compared to the "Standard DDIM".

| Model | Method | Extra Model | FVD ($\downarrow$) (StyleGAN) | FVD ($\downarrow$) (VideoGPT) |
|---|---|---|---|---|
| Animatediff (SD V1.5, Motion Adapter V3) | Standard DDIM | ✗ | $762.22_{\pm 0.019}$ | $761.04_{\pm 0.041}$ |
| | FreeInit | ✗ | $732.59_{\pm 0.013}$ | $731.05_{\pm 0.022}$ |
| | I4VGen | ✓ | $741.55_{\pm 0.019}$ | $740.11_{\pm 0.021}$ |
| | *IV-mixed Sampler* (ours) | ✓ | $721.31_{\pm 0.011}$ ($+\Delta$ 40.91) | $719.92_{\pm 0.001}$ ($+\Delta$ 41.12) |
| ModelScope-T2V | Standard DDIM | ✗ | $739.13_{\pm 0.020}$ | $737.23_{\pm 0.017}$ |
| | *IV-mixed Sampler* (ours) | ✓ | $603.31_{\pm 0.020}$ ($+\Delta$ 135.82) | $601.88_{\pm 0.016}$ ($+\Delta$ 135.35) |

et al., 2023), and VideoCrafterV2 (Chen et al., 2023a). Additionally, we compare *IV-mixed Sampler* with three heavy-inference algorithms, FreeInit (Wu et al., 2023), Unictrl (Chen et al., 2024d) and I4VGen (Guo et al., 2024), on VDM. Note that I4VGen is also an algorithm designed to enhance VDM performance using IDM. However, its IDM is configured to be consistent with VDM, achieving image synthesis by reshaping from b×c×t×h×w to (b×t)×c×1×h×w before passing it through VDM. For further details, please refer to Appendix A.3. Finally, we conduct extensive ablation studies and present visualization to validate the optimal solution of various design choices. More implementation details can be found in Appendix A.

## 4.1 Main Results

**Chronomagic-Bench 150 & 1649.** We evaluate the effectiveness of *IV-mixed Sampler* on three different VDMs: VideoCrafterV2, ModelScope-T2V, and Animatediff (SD V1.5, Motion Adapter V3). The comparison results on Chronomagic-Bench-150 (*w.r.t*, 150 prompts) are presented in Table 1. We employ three metrics in this benchmark: UMT-FVD (for visual quality), UMTScore (for semantic faithfulness), and GPT4o-MTScore (for temporal coherence and metamorphic amplitude) to assess our proposed method. The experimental results in Table 1 show that *IV-mixed Sampler* significantly outperforms both the standard sampling method and other computationally intensive algorithms. This indicates that rational integration of IDM and VDM has great potential to improve the performance of video synthesis tasks. Although *IV-mixed Sampler* does not outperform all comparative methods on Animatediff and ModelScope-T2V in terms of UMT-FVD, it still shows a significant improvement over standard sampling algorithms. It is worth noting that ModelScope-T2V was unable to locate a high-quality IDM due to its low resolution (*i.e.*, 224×224) synthesized video. In contrast, *IV-mixed Sampler* do not perform best on Animatediff because I4VGen is an algorithm that integrates both IDM and VDM. Moreover, on the more comprehensive Chronomagic-Bench-1649 (*w.r.t*, 1649 prompts), *IV-mixed Sampler* outperforms all comparative methods across all metrics and models. As illustrated in Table 2, *IV-mixed Sampler* achieves the best performance on all metrics, even when compared to the latest SOTA method, I4VGen. These experimental results underscore the strong generalization capabilities of *IV-mixed Sampler*, highlighting its effectiveness as a plug-and-play solution that can be seamlessly integrated across various VDMs.

**VBench.** For a comprehensive evaluation of both visual quality and semantic consistency, we further assessed the performance of *IV-mixed Sampler* on VBench (Huang et al., 2024), with the results presented in Table 5. From Table 5, it is clear that *IV-mixed Sampler* outperforms vanilla sampling across most metrics, particularly on the multiple objects, where *IV-mixed Sampler* improved the performance of Animatediff from 36.88% to 58.46%. Additionally, the average scores of *IV-*

Table 5: **Quantitative comparison on VBench (Huang et al., 2024).** $(+\Delta [\cdot])$ represents the improvement of *IV-mixed Sampler* compared to Vanilla Sampling.

| Model | Method | Average Score | Subject Consistency | Temporal Flickering | Object Class | Multiple Objects | Human Action |
|---|---|---|---|---|---|---|---|
| Animatediff (SD V1.5, Motion Adapter V2) | Vanilla Sampling | 60.19% | 95.30% | 98.75% | 90.90% | 36.88% | 92.60% |
| | IV-mixed Sampler | 66.69% (+△ 6.50%) | 93.31% | 97.09% | 96.50% (+△ 5.60%) | 58.46% (+△ 21.58%) | 98.60% (+△ 6.00%) |
| Modelscope-T2V | Vanilla Sampling | 57.35% | 89.87% | 98.28% | 82.25% | 38.98% | 92.40% |
| | IV-mixed Sampler | 57.89% (+△ 0.54%) | 88.39% | 98.98% (+△ 0.70%) | 79.42% | 38.05% | 94.60% (+△ 2.20%) |
| Model | Method | Color | Spatial Relationship | Scene | Appearance Style | Temporal Style | Overall Consistency |
| Animatediff (SD V1.5, Motion Adapter V2) | Vanilla Sampling | 87.47% | 34.60% | 50.19% | 22.42% | 26.03% | 27.04% |
| | IV-mixed Sampler | 91.30% (+△ +3.83%) | 59.78% (+△ +25.18%) | 56.91% (+△ 6.72%) | 24.44% (+△ 2.02%) | 27.62% (+△ 1.59%) | 29.54% (+△ 2.50%) |
| Modelscope-T2V | Vanilla Sampling | 81.72% | 33.68% | 39.26% | 23.39% | 25.37% | 25.67% |
| | IV-mixed Sampler | 83.96% (+△ 2.24%) | 35.42% (+△ +1.74%) | 42.62% (+△ 3.36%) | 23.51% (+△ 0.12%) | 25.44% (+△ 0.07%) | 26.43% (+△ 0.76%) |

*mixed Sampler* exceeded those of vanilla sampling on both Modelscope-T2V and Animatediff, fully demonstrating the effectiveness of *IV-mixed Sampler*.

**UCF-101-FVD & MSR-VTT-FVD.** Additionally, to complement our experimental results, we conduct experiments on several traditional benchmarks, including UCF-101-FVD and MSR-VTT-FVD. The FVD results for UCF-101 and MSR-VTT are presented in Table 3 and Table 4, respectively. These quantitative results significantly substantiate the superiority of *IV-mixed Sampler*, highlighting its effectiveness in outperforming existing heavy-inference approaches on VDMs. Through these results and visualized in Fig. 1, *IV-mixed Sampler* has strong generalization ability across different VDMs, which possesses significant practical application value in real-world scenarios.

## 4.2 ABLATION STUDIES

We begin with a basic ablation study to demonstrate the validity of inserting IDM into the denoising process of VDM. We present the experimental outcomes in Fig. 6, we can discover that the approach "R-[·]", which use Gaussian noise to perform the forward diffusion process, result in significantly lower quality of the synthesized video compared to the standard DDIM process (*i.e.*, Origin in Fig. 6). This phenomenon arises because "R-[·]" over-introduces invalid information (*i.e.*, Gaussian noise) into the synthesized video during denoising. Thus, we consider the more robust DDIM-Inversion to integrate the video denoising process with the image denoising process, as this paradigm is stable and effectively reduces truncation errors in practical discrete sampling.

After that, as described in Sec. 3.3, we elucidate three design choices, namely the sampling interval of *IV-mixed Sampler*, the $C_4^2$ species combinations, and the dynamic CFG scale. For *1) the sampling interval of IV-mixed Sampler*, the results in Fig. 3 clearly illustrate that performing *IV-mixed Sampler* across all sampling steps is optimal. Another evident conclusion is that the closer the sampling interval of *IV-mixed Sampler* is to $t=1$, the more significant the performance gain. This suggests that to save computational overhead, *IV-mixed Sampler* can be applied within the 0%-50% sampling interval or even restricted to the 0%-25% interval. For *2) the $C_4^2$ species combinations*, we present its ablation results in Fig. 4. It is evident that "IV-IV" outperforms all metrics and significantly surpasses both FreeInit and standard DDIM. This suggests that there is an empirically optimal combination of results within *IV-mixed Sampler*. Accordingly, we use "IV-IV" for all comparison experiments. For *3) the dynamic CFG scale*, the conclusions are not as intuitive as the first two design choices. Specifically, we considered a total of 7 combinations of

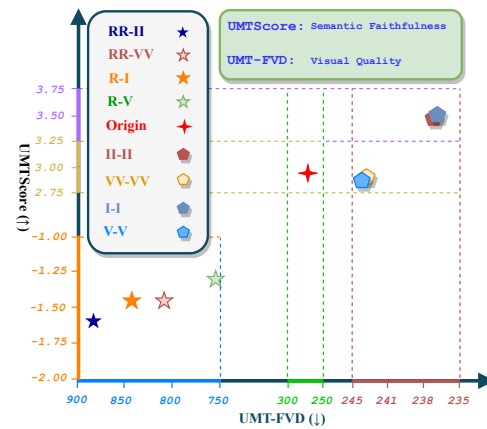

Figure 6: UMTScore (↑) vs. UMT-FVD (↓) with Animatediff (Guo et al., 2023) on Chronomagic-Bench-150 (Yuan et al., 2024). In the legend, "R", "I", and "V" represent the score function estimation using random Gaussian noise, IDM, and VDM, respectively. Moreover, the front of the horizontal line "-" refers to the additive noise form, while the back of "-" represents the denoising paradigm. For instance, "RR-II" stands for a two-step of adding noise with Guassian noise followed by two-step of denoising performed using IDM.

dynamic CFG scales, where the CFG scale varies from $\gamma_{\text{go}}^{\text{t=0}}$ to $\gamma_{\text{go}}^{\text{t=1}}$ across IDM dynamic cfg scale, VDM dynamic cfg scale and $\rho$. In Fig. 5, we set $\rho$ to 7, 1, and $1/7$[1] to model changes in the CFG scale as convex, straight, and concave functions, respectively. For the specified $\rho = 1$, we consider 3 types of the IDM's CFG scale $\omega_{\text{go-back}}^{\text{IDM}}$ and VDM's CFG scale $\omega_{\text{go-back}}^{\text{VDM}}$: (1) remaining constant, (2) $\omega_{\text{go-back}}^{\text{IDM}}/\omega_{\text{go-back}}^{\text{VDM}}$ increasing, (3) $\omega_{\text{go-back}}^{\text{IDM}}/\omega_{\text{go-back}}^{\text{VDM}}$ decreasing. For the "constant" case we make $\gamma_{\text{go}}^{\text{t=0}} = \gamma_{\text{go}}^{\text{t=1}} = 4$, for the "decreasing" case we make $\gamma_{\text{go}}^{\text{t=0}} = 6$, $\gamma_{\text{go}}^{\text{t=1}} = 2$, and for the "increasing" we make $\gamma_{\text{go}}^{\text{t=0}} = 2$, $\gamma_{\text{go}}^{\text{t=1}} = 6$. As illustrated in Fig. 5, we find that for visual quality (*w.r.t.*, UMT-FVD), keeping both $\omega_{\text{go-back}}^{\text{IDM}}$ and $\omega_{\text{go-back}}^{\text{VDM}}$ constant is the best choice. For semantic faithfulness (*w.r.t.*, UMTScore), the optimal strategy is to have both $\omega_{\text{go-back}}^{\text{IDM}}$ and $\omega_{\text{go-back}}^{\text{VDM}}$ increasing. For temporal coherence (*w.r.t.*, GPT4o-MTScore), it is optimal for $\omega_{\text{go-back}}^{\text{IDM}}$ to decrease while $\omega_{\text{go-back}}^{\text{VDM}}$ increases. Therefore, to enhance the performance of a specific aspect of synthesized video, we can adjust the dynamic CFG scale to achieve the empirically optimal trade-off.

**Visualization.** We visualize the standard sampling and *IV-mixed Sampler* of the synthesized video in Fig. 1. It can be observed that *IV-mixed Sampler* significantly improves both visual quality and semantic faithfulness. In addition to this, we empirically invited a number of other AIGC-related researchers to judge the video quality and agreed that *IV-mixed Sampler*'s enhancement could be observed by the naked eye.

## 5 LIMITATION

Althrough *IV-mixed Sampler* significantly improves the performance of VDM, it introduces additional computational costs. For "IV-IV" on Animatediff, it increases the number of function evaluation (NFE) from 50 to 250. In the practical implementation, the computational overhead went up from 21s to 92s at a single RTX 4090 GPU. This problem could potentially be addressed in the future by distillation algorithms similar to accelerated sampling (Song et al., 2023b; Shao et al., 2023; Salimans & Ho, 2022). This exploration of INFERENCE SCALING LAWS first, and then distilling the performance gains it achieves back to the foundation model may be a viable path for the future.

## 6 CONCLUSION

In this paper, we propose *IV-mixed Sampler* to enhance the visual quality of synthesized videos by leveraging an IDM while ensuring temporal coherence through a VDM. The algorithm utilizes DDIM and DDIM-Inversion to correct latent representations $\mathbf{x}_t$ at any time point $t$, enabling seamless integration into any VDM and sampling interval. *IV-mixed Sampler* can be formulated as an ODE, achieving a trade-off between visual quality and temporal coherence by adjusting the CFG scales of both the IDM and VDM. In the future, we plan to fine-tune several stronger IDMs, such as FLUX, to better adapt the latent space of target VDMs, thereby further enhancing the performance of VDMs. We anticipate *IV-mixed Sampler* will be widely applicable in vision generation tasks.

## ACKNOWLEDGE

This work was supported by Guangdong Provincial Key Lab of Integrated Communication, Sensing and Computation for Ubiquitous Internet of Things (No. 2023B1212010007).

---

[1] $\rho = 7$ is the default implementation of Karras's noise schedule (Karras et al., 2022).

**Ethics Statement.** We present *IV-mixed Sampler*, a method designed to enhance the semantic accuracy and visual quality of video produced by Video Diffusion Models. Although our approach does not directly engage with real-world datasets, we are dedicated to ensuring the ethical use of prompts, while respecting user autonomy and striving for positive outcomes. Acknowledging the commercial potential of *IV-mixed Sampler*, we emphasize a responsible and ethical deployment of the technology, aiming to maximize societal benefits while carefully mitigating any potential risks.

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

# A  ADDITIONAL IMPLEMENTATION DETAILS

## A.1  BENCHMARKS

We present the relevant metrics and benchmarks used for comparison in the main paper..

**UCF-101-related FVD.**  The UCF-101 dataset is an action recognition dataset comprising 101 categories, with all videos sourced from Youtube. Each video has a fixed frame rate of 25 frames per second (FPS) and a resolution of 320×240. Several previous works (Blattmann et al., 2023; Wu et al., 2023; Chen et al., 2024d) have validated the generation performance of VDMs on the UCF-101 dataset using Fréchet Video Distance (FVD) (Unterthiner et al., 2019). However, a comprehensive evaluation benchmark for UCF-101 is still lacking. To address this, we follow the methodology of FreeInit, utilizing the prompts listed in Ge et al. (2023) to synthesize videos and assess inference performance with FVD. Specifically, we synthesize 5 videos for each of the 101 prompts provided by Ge et al. (2023), resulting in a total of 505 synthesized videos. We then compute the FVD between these 505 synthesized videos and 505 randomly sampled videos from the UCF-101 dataset (5 per class), using the built-in FVD evaluation code from Open-Sora-Plan[2].

**MSR-VTT-related FVD.**  The MSR-VTT dataset (Xu et al., 2016) is a large-scale dataset for open-domain video captioning, featuring 10,000 video clips categorized into 20 classes. The standard split of the MSR-VTT dataset includes 6,513 clips for training, 497 clips for validation, and 2,990 clips for testing. For our evaluation, we utilize all 497 validation videos. To ensure evaluation stability, we synthesize a total of 1,491 videos based on prompts from these validation videos, with each prompt producing 3 different videos. We assess the results using the built-in FVD evaluation code from Open-Sora-Plan.

**Chronomagic-Bench-150.**  Chronomagic-Bench-150, introduced in (Yuan et al., 2024) and recently accepted by NeurIPS 2024's dataset and benchmark track, serves as a comprehensive benchmark for metamorphic evaluation of timelapse T2V synthesis. This benchmark includes 4 main categories of time-lapse videos: biological, human-created, meteorological, and physical, further divided into 75 subcategories. Each subcategory contains two challenging prompts, leading to in a total of 150 prompts. We consider three distinct metrics in Chronomagic-Bench-150: UMT-FVD ($\downarrow$), UMTScore ($\uparrow$), and GPT4o-MTScore ($\uparrow$), each addressing different evaluation aspects. Specifically, UMT-FVD ($\downarrow$) (Liu et al., 2024b) leverages the UMT (Li et al., 2023) feature space to compute FVD, assessing the visual quality of the synthesized video. UMTScore ($\uparrow$) utilizes the UMT (Li et al., 2023) feature space to compute CLIPScore (Hessel et al., 2021), evaluating the text relevance of the synthesized video. Lastly, GPT4o-MTScore ($\uparrow$) is a fine-grained metric that employs GPT-4o (Achiam et al., 2023) as an evaluator, aligning with human perception to accurately reflect the metamorphic amplitude and temporal coherence of T2V models.

**Chronomagic-Bench-1649.**  Chronomagic-Bench-1649, introduced in (Yuan et al., 2024) and recently accepted by NeurIPS 2024's dataset and benchmark track, is a comprehensive benchmark designed for the metamorphic evaluation of timelapse T2V synthesis. While it shares 75 subcategories with Chronomagic-Bench-150, it offers a more extensive evaluation framework with 1649 prompts, making it significantly more comprehensive than its lightweight counterpart Chronomagic-Bench-150. Chronomagic-Bench-1649 includes 4 key metrics: UMT-FVD ($\downarrow$), MTScore ($\uparrow$), UMTScore ($\uparrow$), and GPT4o-MTScore ($\uparrow$), each serving to evaluate different aspects of video synthesis. Specifically, UMT-FVD ($\downarrow$) (Liu et al., 2024b) utilizes the UMT (Li et al., 2023) feature space to compute FVD, assessing the visual quality of the synthesized videos. MTScore ($\uparrow$) measures metamorphic amplitude, indicating the degree of change between frames. UMTScore ($\uparrow$) leverages the UMT (Li et al., 2023) feature space to compute CLIPScore (Hessel et al., 2021), evaluating the text relevance of the synthesized videos. Finally, GPT4o-MTScore ($\uparrow$) is a fine-grained metric that employs GPT-4o (Achiam et al., 2023) as an evaluator, aligning with human perception to accurately reflect the metamorphic amplitude and temporal coherence of T2V models. As with to Chronomagic-Bench-150, we choose to ignore the MTScore ($\uparrow$) metric in our experiments due to its limitations.

---

[2] https://github.com/PKU-YuanGroup/Open-Sora-Plan

## A.2 Video Diffusion Models

We describe the VDMs utilized in this work. Specifically, we employ 3 VDMs with distinct architectures: ModelScope-T2V (Wang et al., 2023), Animatediff (Guo et al., 2023), VideoCrafterV2 (Chen et al., 2024a).

**ModelScope-T2V.** ModelScope-T2V incorporates spatio-temporal blocks to ensure consistent frame generation and smooth motion transitions. Its key features include the utilization of 3D convolution andtraining from scratch. The input video size is structured as $3 \times 16 \times 256 \times 256$, where 3 represents the number of channels, 16 is the number of frames, and 256×256 indicates the resolution of each frame. This configuration allows the model to effectively capture both spatial adn temporal features, facilitating high-quality video synthesis.

**Animatediff.** Animatediff does not require training from scratch; instead, it only needs fine-tuning on existing image diffusion models. Its motion adapter serves as a plug-and-play module, allowing most community text-to-image models to be transformed into animation generators. In this paper, we consider the latest version Animatediff (SD V1.5, Motion Adapter V3), which is fine-tuned from SD V1.5. The input video size of Animatediff (SD V1.5, Motion Adapter V3) is $3 \times 16 \times 512 \times 512$, where 3 indicates the number of channels, 16 represents the number of frames, and 512×512 specifies the resolution. Note that there are differences in the performance of this VDM because we used a different resolution than the one used in the Chronomagic-Bench paper (Yuan et al., 2024).

**VideoCrafterV2.** VideoCrafterV2 focuses on T2V synthesis, aiming to synthesize high-quality videos from prompts. This work investigates a training scheme for video models based on Stable Diffusion (Rombach et al., 2022b), exploring how to leverage low-quality videos and synthesized high-quality images to develop a superior video model. The input video size of VideoCrafterV2 is $3 \times 16 \times 512 \times 320$, where 3 indicates the number of channels, 16 represents the number of frames, and $512 \times 320$ specifies the resolution.

## A.3 Heavy-inference Algorithm on VDM

Here we discuss two popular VDM-based heavy-inference algorithms FreeInit (Wu et al., 2023) and I4VGen (Guo et al., 2024).

**I4VGen.** I4VGen (Guo et al., 2024) is a training-free and plug-and-play video diffusion inference framework that enhances text-to-video synthesis by leveraging robust image techniques. To be specific, I4VGen decomposes the process into two stages: anchor image synthesis and anchor image-guided video synthesis. A well-designed generation-selection pipeline is used to create visually realistic and semantically faithful anchor images, while score distillation sampling (SDS) (Poole et al., 2023) is employed to animate the images into dynamic videos, followed by a video regeneration process to refine the output. In its official implementation, both phases are realized by VDM, where the anchor image synthesis is performed by merging the time dimension into the batch size dimension through VDM. In essence, I4VGen does not introduce true IDMs to improve the quality of synthesized video obtained from VDMs.

**FreeInit.** FreeInit (Wu et al., 2023) is a novel inference-time strategy designed to enhance temporal consistency in video generation using diffusion models. This approach addresses a key issue: the difference in the spatial-temporal frequency distribution of noise between training and inference, which leads to poor video quality. FreeInit iteratively refines the low-frequency components of the initial noise during inference, bridging this gap without requiring additional training.

## A.4 Hyperparameter Settings

For all comparison experiments, we used the form "IV-IV" and perform *IV-mixed Sampler* at all time steps of the standard DDIM sampling. In addition, $\gamma_{go}^{t=0}$, $\gamma_{back}^{t=0}$, $\gamma_{go}^{t=1}$ and $\gamma_{back}^{t=1}$ all are set as 4. For both Animatediff and ModelScope-T2V, we use stable diffusion (SD) V1.5 as the IDM. Note that we experimented with using Mini SD as the IDM for ModelScope-T2V to maintain a consistent resolution of 256×256. However, as illustrated in Table 6, we found that its performance was

Table 6: **Ablation studies with Modelscope-T2V on Chronomagic-Bench-150.**

| Model | Method | Extra Model | UMT-FVD ($\downarrow$) | UMTScore ($\uparrow$) | GPT4o-MTScore ($\uparrow$) |
|---|---|---|---|---|---|
| | Standard DDIM | ✗ | 241.61 | 2.66 | 2.96 |
| ModelScope-T2V | *IV-mixed Sampler* (Mini SD) | ✓ | 247.99 | 2.63 | 3.00 |
| | *IV-mixed Sampler* (SD V1.5) | ✓ | 234.90 ($+\triangle$ 6.71) | 3.02 ($+\triangle$ 0.36) | 3.14 ($+\triangle$ 0.18) |

Table 7: **Performance comparison of VideoCrafterV2 across different $z\%$ settings.**

| Model | $z\%$ | UMT-FVD ($\downarrow$) | UMTScore ($\uparrow$) | GPT4o-MTScore ($\uparrow$) |
|---|---|---|---|---|
| | Standard DDIM | 214.06 | 2.76 | 2.87 |
| | 33.3% | 212.74 | 3.08 | 3.02 |
| VideoCrafterV2 | 50.0% | 208.67 ($+\triangle$ 5.39) | 3.23 | 3.18 |
| | 66.7% | 210.57 | 3.29 ($+\triangle$ 0.53) | 3.30 ($+\triangle$ 0.43) |
| | 75.0% | 211.87 | 3.28 | 3.28 |

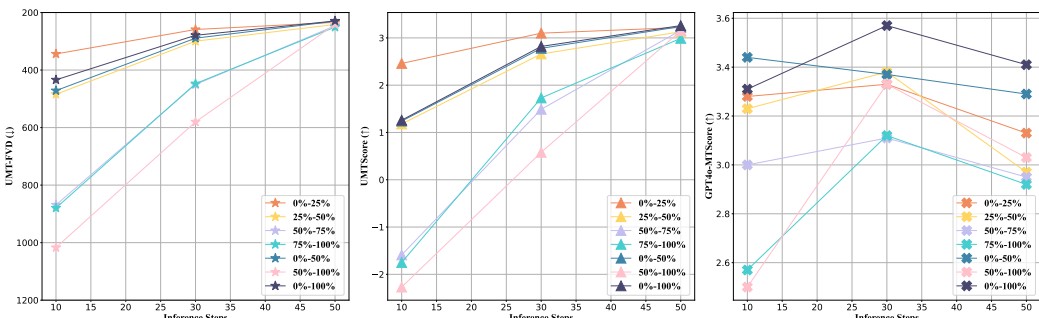

Figure 7: Ablation studies on sampling intervals of *IV-mixed Sampler* ("VI-IV"). The BEGIN%-END% in the legend indicates the portion of the entire sampling process performed by *IV-mixed Sampler*.

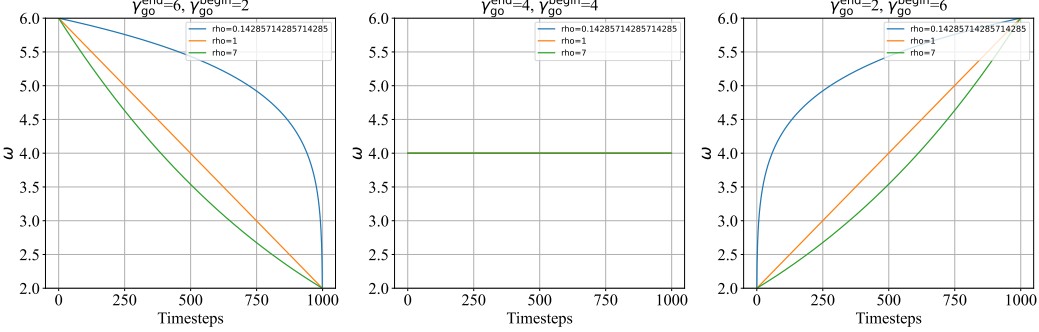

Figure 8: The visualization of Karras's noise schedule.

inferior to using SD V1.5 with upsampling and downsampling. For VideoCrafterV2, we use Realistic Vision V6.0 B1 (Mage.Space, 2023) as the IDM to accommodate a resolution of 512×320. For the remaining configurations, we follow the sampling form recommended by the corresponding VDMs. Furthermore, we find that applying "IV-IV" at every step on VideoCrafterV2 destroys temporal coherence. Therefore, we replace "IV-IV" with "VV-VV" for z%. The results of the ablation experiments are shown in Table 7. We finally chose z%=66.7% as the final solution.

## B  PYTHON-STYLE PSEUDO CODE

We present the python-style pseudo-code in Fig. 1 to make it easier to understand *IV-mixed Sampler*.

**Algorithm 1** Pseudo code of *IV-mixed Sampler* (*w.r.t.*, "IV-IV") (⤸) in a PyTorch-like style.

```python
# x_T: the initial noise
# N: the number of sampling step
# scheduler: a scheduler similar to that of diffusers
# inv_scheduler: an inversion scheduler similar to that of diffusers
# F_idm, F_vdm: the backbones of IDM and VDM
# pe_idm, pe_vdm: the prompt embedding of IDM and VDM
# w_idm_begin, w_idm_end, w_vdm_begin, w_vdm_end, rho:
#    the hyperparameter of the dynamic CFG scale

def get_sigmas_karras(n, sigma_begin, sigma_end, rho=7.0):
        ramp = torch.linspace(0, 1, n)
        begin_inv_rho = sigma_begin ** (1 / rho)
        end_inv_rho = sigma_end ** (1 / rho)
        sigmas = (begin_inv_rho +
          ramp * (end_inv_rho - begin_inv_rho)) ** rho
        return sigmas # Constructs the noise schedule of Karras et al.

def sampling(latent, scheduler, inv_scheduler, F_idm, F_vdm, **kwargs):
        pe_idm, pe_vdm = kwargs.get("pe_idm", None), kwargs.get("pe_vdm", None)
        w_idm_begin, w_idm_end, w_vdm_begin, w_vdm_end, rho =
          kwargs.get("w_idm_begin", 4), kwargs.get("w_idm_end", 4),
          kwargs.get("w_vdm_begin", 4), kwargs.get("w_vdm_end", 4),
          kwargs.get("rho", 1)
        N = kwargs.get("N", 50)
        idm_CFG = get_sigmas_karras(N, w_idm_begin, w_idm_end, rho) # Dynamic CFG scale
        vdm_CFG = get_sigmas_karras(N, w_vdm_begin, w_vdm_end, rho)

        for step in range(N):
            t = scheduler.timesteps[step]
            p_t = min(t + scheduler.config.num_train_timesteps
               // scheduler.num_inference_steps,
               scheduler.config.num_train_timesteps-1)
            pp_t = min(t + 2 * scheduler.config.num_train_timesteps
               // scheduler.num_inference_steps,
               scheduler.config.num_train_timesteps-1)
            # Perform sampling with IDM
            latent = einops.rearrange(latent, "b c t h w -> (b t) c h w")
            n_latent =  torch.cat([latent] * 2)
            n_latent = scheduler.scale_model_input(n_latent, t)
            n_pred = F_idm(n_latent, t, pe_idm)
            n_pred_un, n_pred_te = n_pred.chunk(2)
            n_pred = n_pred_un + idm_CFG[step] * (n_pred_te - n_pred_un)
            latent = inv_scheduler.step(n_pred, p_t, latent)
            latent = einops.rearrange(latent, "(b t) c h w -> b c t h w", t=(...) )
            # Perform sampling with VDM
            n_latent =  torch.cat([latent] * 2)
            n_latent = scheduler.scale_model_input(n_latent, p_t)
            n_pred = F_vdm(n_latent, p_t, pe_vdm)
            n_pred_un, n_pred_te = n_pred.chunk(2)
            n_pred = n_pred_un + vdm_CFG[step] * (n_pred_te - n_pred_un)
            latent = inv_scheduler.step(n_pred, pp_t, latent)
            # Perform sampling with IDM
            ...
            n_latent = scheduler.scale_model_input(n_latent, pp_t)
            ...
            latent = scheduler.step(n_pred, pp_t, latent)
            # Perform sampling with VDM
            ...
            n_latent = scheduler.scale_model_input(n_latent, p_t)
            ...
            latent = scheduler.step(n_pred, p_t, latent)
            # Perform sampling with VDM
            ...
            n_latent = scheduler.scale_model_input(n_latent, t)
            ...
            latent = scheduler.step(n_pred, t, latent)
        return decode_latent(latent)          # transfer latent to video
# IV-mixed Sampler
video = sampling(x_T, scheduler, inv_scheduler, ... )
```

## C  THEORETICAL PROOF

Here we give the proof of Theorem 3.1 in the main paper. We use "IV-IV" for an example, and the derivation of other forms of *IV-mixed Sampler* is similar to "IV-IV" and is not described additionally. First, *IV-mixed Sampler* (*w.r.t.*, "IV-IV") can be rewritten as

$$
\begin{aligned}
\mathbf{x}'_t = \mathbf{x}_t &+ \omega_{\text{go}}^1 \frac{g^2(t)}{2} \nabla_{\mathbf{x}} \log q_t^{\text{IDM}}(\mathbf{c}|\mathbf{x}) + \omega_{\text{go}}^2 \frac{g^2(t+\eta)}{2} \nabla_{\mathbf{x}} \log q_{t+\eta}^{\text{VDM}}(\mathbf{c}|\mathbf{x}) \\
&- \omega_{\text{back}}^2 \frac{g^2(t+\eta)}{2} \nabla_{\mathbf{x}} \log q_{t+\eta}^{\text{IDM}}(\mathbf{c}|\mathbf{x}) - \omega_{\text{back}}^1 \frac{g^2(t)}{2} \nabla_{\mathbf{x}} \log q_t^{\text{VDM}}(\mathbf{c}|\mathbf{x}),
\end{aligned}
$$

(11)

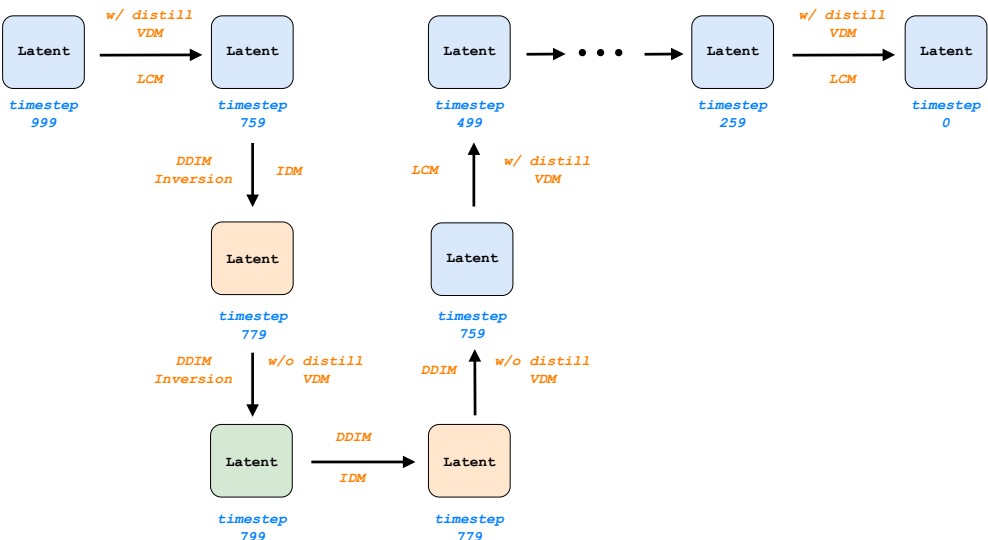

Figure 9: **Overview of our *IV-mixed Sampler* on the motion consistency model (Zhai et al., 2024).**

Table 8: **Quantitative comparison of motion consistency model (MCM) on Chronomagic-Bench-150.** $(+\Delta [\cdot])$ stands for the improvement of *IV-mixed Sampler* compared to the "Standard LCM".

| Model | Method | Extra Model | UMT-FVD ($\downarrow$) | UMTScore ($\uparrow$) | GPT4o-MTScore ($\uparrow$) |
|---|---|---|---|---|---|
| MCM | Standard LCM | ✗ | $277.91_{\pm 0.025}$ | $1.98_{\pm 0.012}$ | $2.66_{\pm 0.021}$ |
| (Animatediff V2) | *IV-mixed Sampler* (ours) | ✓ | $246.59_{\pm 0.021} \ (+\Delta\ 31.32)$ | $2.19_{\pm 0.013} \ (+\Delta\ 0.21)$ | $2.87_{\pm 0.009} \ (+\Delta\ 0.21)$ |

where $\eta$ represents the sampling step, which is usually extremely small. $\nabla_{\mathbf{x}} \log q_t^{\text{IDM}}(\mathbf{c}|\mathbf{x})$ and $\nabla_{\mathbf{x}} \log q_t^{\text{VDM}}(\mathbf{c}|\mathbf{x})$ represent the score function estimated for $p_t(\mathbf{x}_t)$ using IDM and VDM under classifier-free guidance. Since $\eta$ is very small, and assume that $\omega_{\text{back}}^2$ is larger than $\omega_{\text{go}}^1$ and $\omega_{\text{back}}^1$ is larger than $\omega_{\text{go}}^2$, Eq. 11 can be rewritten as

$$\mathbf{x}_t' = \mathbf{x}_t - \omega_{\text{go-back}}^{\text{IDM}} \frac{g^2(t)}{2} \nabla_{\mathbf{x}} \log q_t^{\text{IDM}}(\mathbf{c}|\mathbf{x}) - \omega_{\text{go-back}}^{\text{VDM}} \frac{g^2(t)}{2} \nabla_{\mathbf{x}} \log q_t^{\text{VDM}}(\mathbf{c}|\mathbf{x}), \tag{12}$$

where $\omega_{\text{go-back}}^{\text{IDM}}$ and $\omega_{\text{go-back}}^{\text{VDM}}$ are large than $0$. Given this, *IV-mixed Sampler* can be written as an ordinary differential equation:

$$d\mathbf{x}_t = f(t)\mathbf{x}_t - \frac{1}{2}g^2(t) \left[ \omega_{\text{go-back}}^{\text{IDM}} \frac{g^2(t)}{2} \nabla_{\mathbf{x}} \log q_t^{\text{IDM}}(\mathbf{c}|\mathbf{x}) + (\omega_{\text{go-back}}^{\text{VDM}} + \omega) \frac{g^2(t)}{2} \nabla_{\mathbf{x}} \log q_t^{\text{VDM}}(\mathbf{c}|\mathbf{x}), \right], \tag{13}$$

where $\omega$ refers to the vanilla CFG scale.

## D  ADDITIONAL ABLATION STUDY

We present Fig. 7 here as a supplement to Fig 3 (*w.r.t.*, the sampling interval of *IV-mixed Sampler*) in the main paper. As illuatrated in Fig. 3 and Fig. 7, it can be noticed that "IV-IV" performs significantly better than "VI-IV" under almost all settings. Furthermore, we visualize Karras's noise schedule of our proposed dynamic CFG scale in Fig. 8 for clear understanding.

## E  CAN *IV-mixed Sampler* WORK ON CONSISTENCY MODEL?

To verify the effectiveness of *IV-mixed Sampler* on the distillation-based accelerated sampling model, we apply *IV-mixed Sampler* to the motion consistency model (MCM) (Zhai et al., 2024). We use the `Animatediff-laion` version from the official MCM library and set the number of

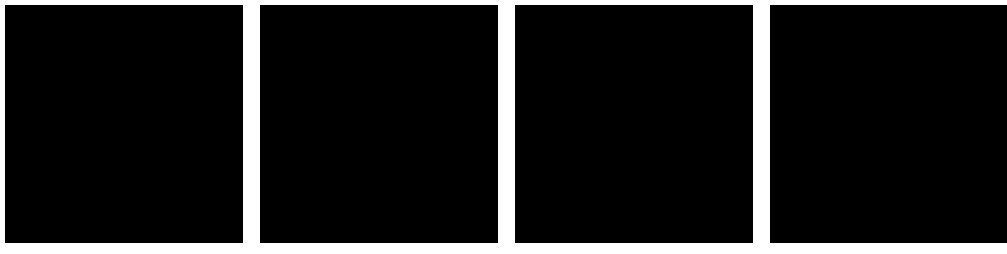

**1th Frame**     **5th Frame**     **9th Frame**     **13th Frame**

Figure 10: **Visualization of our *IV-mixed Sampler* on Animatediff (SD V1.5, Motion Adapter V3) using SD XL as the IDM.**

sampling steps to 4. After extensive empirical experimentation, we found that the sampling mechanism shown in Fig. 9 can enhance MCM performance by introducing IDM. Specifically, we used three checkpoints: the VDM w/o distillation, the VDM w/ distillation, and the standard IDM. During each sampling step with the VDM w/ distillation, IV-mixed sampling is performed using the VDM w/o distillation and the standard IDM. It is important to note that if the VDM w/ distillation is used for IV-mixed sampling (*i.e.*, replacing the VDM w/o distillation with the VDM w/ distillation), it will not work due to its `LCMScheduler` at each step, which is non-deterministic and would disrupt the semantic information provided by the IDM.

We present the results in Table 8, and we can find that *IV-mixed Sampler* significantly improves the performance of MCM in all metrics.

## F  HOW TO MAKE *IV-mixed Sampler* PERFORM BETTER ON MODELSCOPE-T2V?

The performance of *IV-mixed Sampler* largely depends on the generative capacity of the IDM and the VDM. In the previous version of this study, we used SD V1.5 as the IDM on Modelscope-T2V, but now, we choose RealVis-V6.0-B1 (Mage.Space, 2023), which is less sensitive to resolution, as the IDM. We then adjust the CFG scales $\omega_{\text{go}}$ of the IDM and VDM to 2 ($\omega_{\text{back}} = -2$) and 3 ($\omega_{\text{back}} = -3$), respectively, and present the results in Table 9. From these results, we observe that *IV-mixed Sampler* using RealVis-V6.0-B1 as the IDM outperforms FreeInit across all metrics, which highlights the potential of our algorithm.

## G  WHY CANNOT IDM AND VDM USE DIFFERENT VAES?

In the denoising process of *IV-mixed Sampler*, if the latent spaces of the VAEs in IDM and VDM are different, *IV-mixed Sampler* will not function properly. This is similar to attempting direct image classification with a language model without fine-tuning; their high-dimensional semantic spaces are different and therefore incompatible. We demonstrate this by applying SD XL as the IDM to Animatediff (SD V1.5, Motion Adapter V3), as shown in Fig. 10. As illustrated in Fig. 10, the resulting video frames are entirely black and cannot be recognized by the naked eye.

## H  HOW TO MAKE *IV-mixed Sampler* INDEPENDENT OF VAE TO REALIZE PERFORMANCE ENHANCEMENT OF MOCHI-1-PREVIEW?

We made minor modifications to *IV-mixed Sampler* to eliminate the need for ensuring that the VAE of IDM and VDM are the same. To be specific, the changes made to Mochi-1 are shown in Fig. 11 of the revised version, focusing on converting the video latent $x_t$ to $x_0$ using one-step sampling, enhancing it with the ControlNet tile (SD XL), and adding a certain amount of random noise before passing it through Euler's inversion (similar to DDIM-Inversion). This approach allows the video latent to be converted back to $x_t$, thereby improving video fidelity through IDM.

The visualization results can be found in Figs. 12, 13 and 14, where it can be found that *IV-mixed Sampler* greatly improves the video fidelity. For example, for the prompt "A serene forest clearing

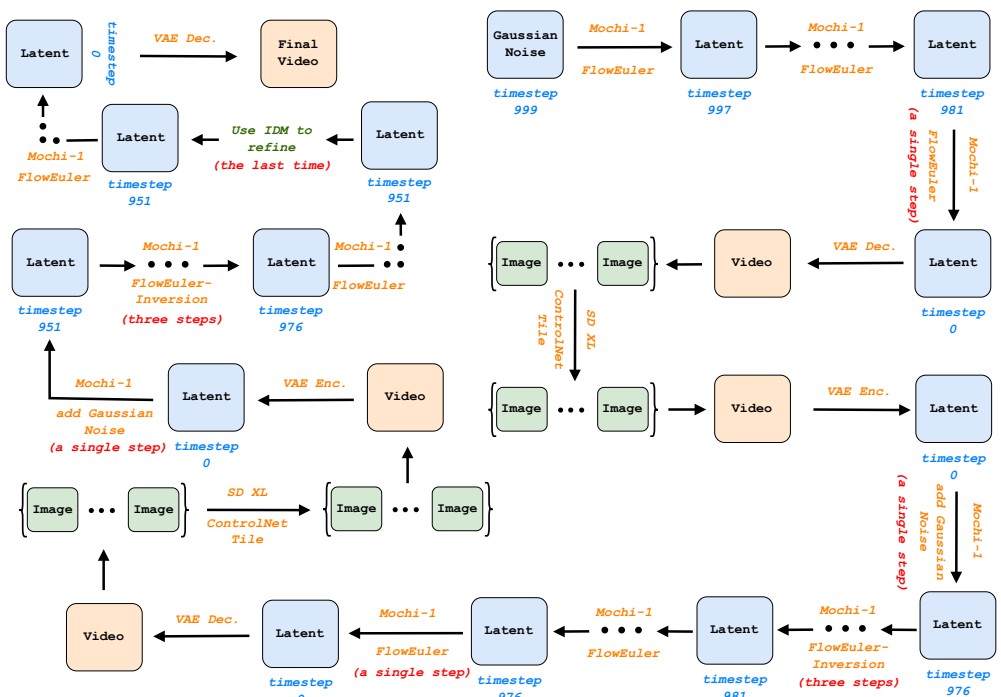

Figure 11: **The illustration of *IV-mixed Sampler* on the state-of-the-art (SOTA) open-source VDM Mochi-1.**

Table 9: **Modelscope-T2V with Better IDM on Chronomagic-Bench-150.**

| Model | Method | Extra Model | UMT-FVD ($\downarrow$) | UMTScore ($\uparrow$) | GPT4o-MTScore ($\uparrow$) |
|---|---|---|---|---|---|
| ModelScope-T2V | Standard DDIM | ✗ | $241.61_{\pm 0.002}$ | $2.65_{\pm 0.017}$ | $2.97_{\pm 0.013}$ |
| | FreeInit | ✗ | $220.96_{\pm 0.002}$ | $2.99_{\pm 0.025}$ | $3.10_{\pm 0.015}$ |
| | *IV-mixed Sampler* (SD V1.5) | ✓ | $234.90_{\pm 0.001}$ $(+\Delta\ 6.71)$ | $3.00_{\pm 0.022}$ $(+\Delta\ 0.35)$ | $3.16_{\pm 0.035}$ $(+\Delta\ 0.19)$ |
| | *IV-mixed Sampler* (RealVis-V6.0-B1) | ✓ | $217.25_{\pm 0.012}$ $(+\Delta\ 24.36)$ | $3.31_{\pm 0.013}$ $(+\Delta\ 0.66)$ | $3.25_{\pm 0.010}$ $(+\Delta\ 0.28)$ |

at dawn, where deer graze peacefully while golden rays of sunlight pierce through the mist-laden trees.", *IV-mixed Sampler* not only improves visual quality but also makes the deer in the video walk very naturally, which is really amazing!

## I  WHEN WOULD *IV-mixed Sampler* PERFORM WORSE THAN VANILLA SAMPLING?

In the certain dimension of VBench, particularly in the domain related to temporal, the performance of *IV-mixed Sampler* is inferior to that of Vanilla Sampling. We believe that *IV-mixed Sampler* improves the overall quality of the composite video by balancing visual quality and temporal coherence. However, as shown in Fig. 15, slight inconsistencies can arise if the magnitude of motion change is too large within a few consecutive frames. The best approach to address this issue is to balance visual quality and temporal coherence based on the hyperparameter $z$, as introduced in Appendix A.4.

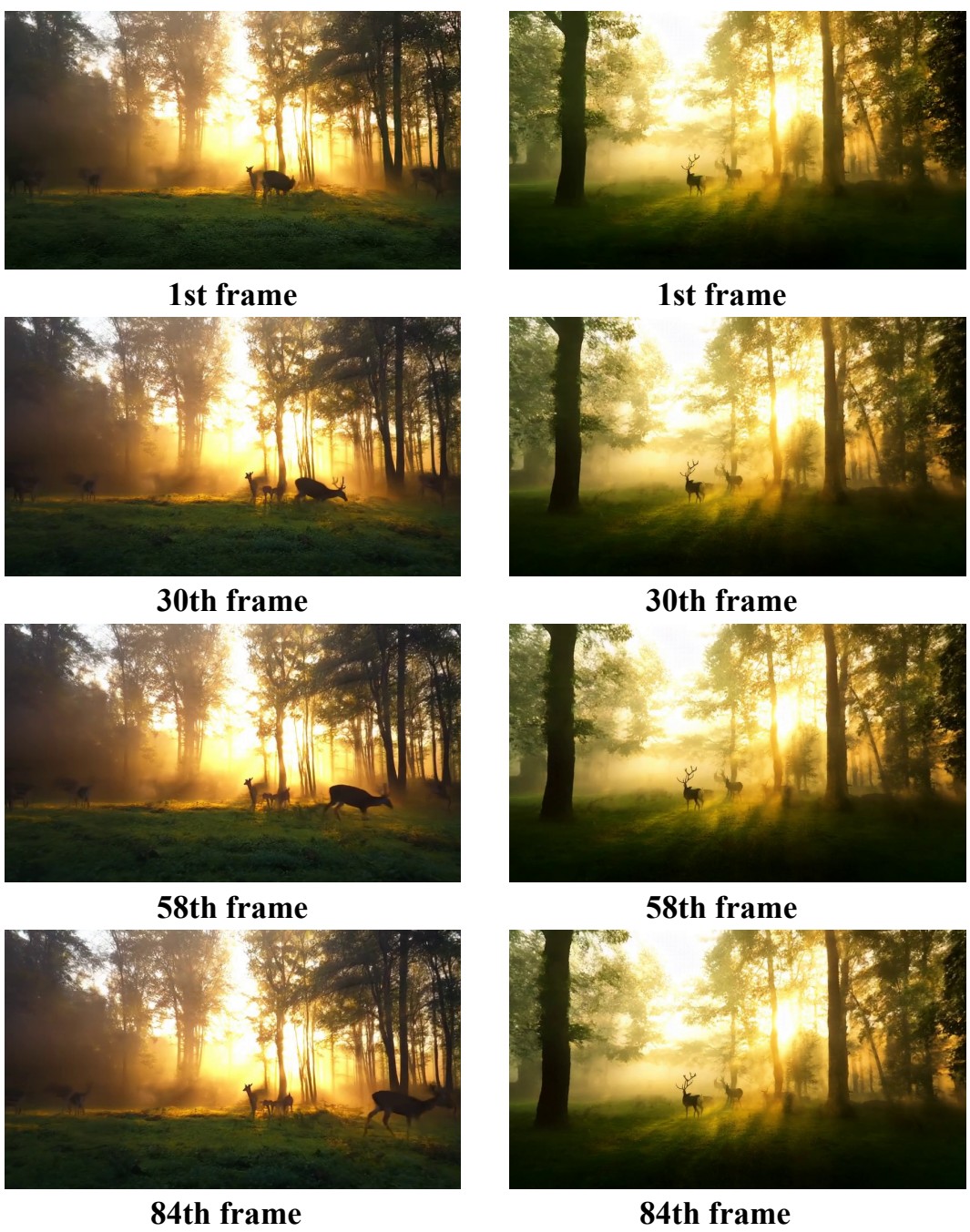

1st frame        1st frame

30th frame        30th frame

58th frame        58th frame

84th frame        84th frame

**A serene forest clearing at dawn, where deer graze peacefully while golden rays of sunlight pierce through the mist-laden trees.**

Figure 12: Left: *IV-mixed Sampler*. Right: Vanilla Sampling.

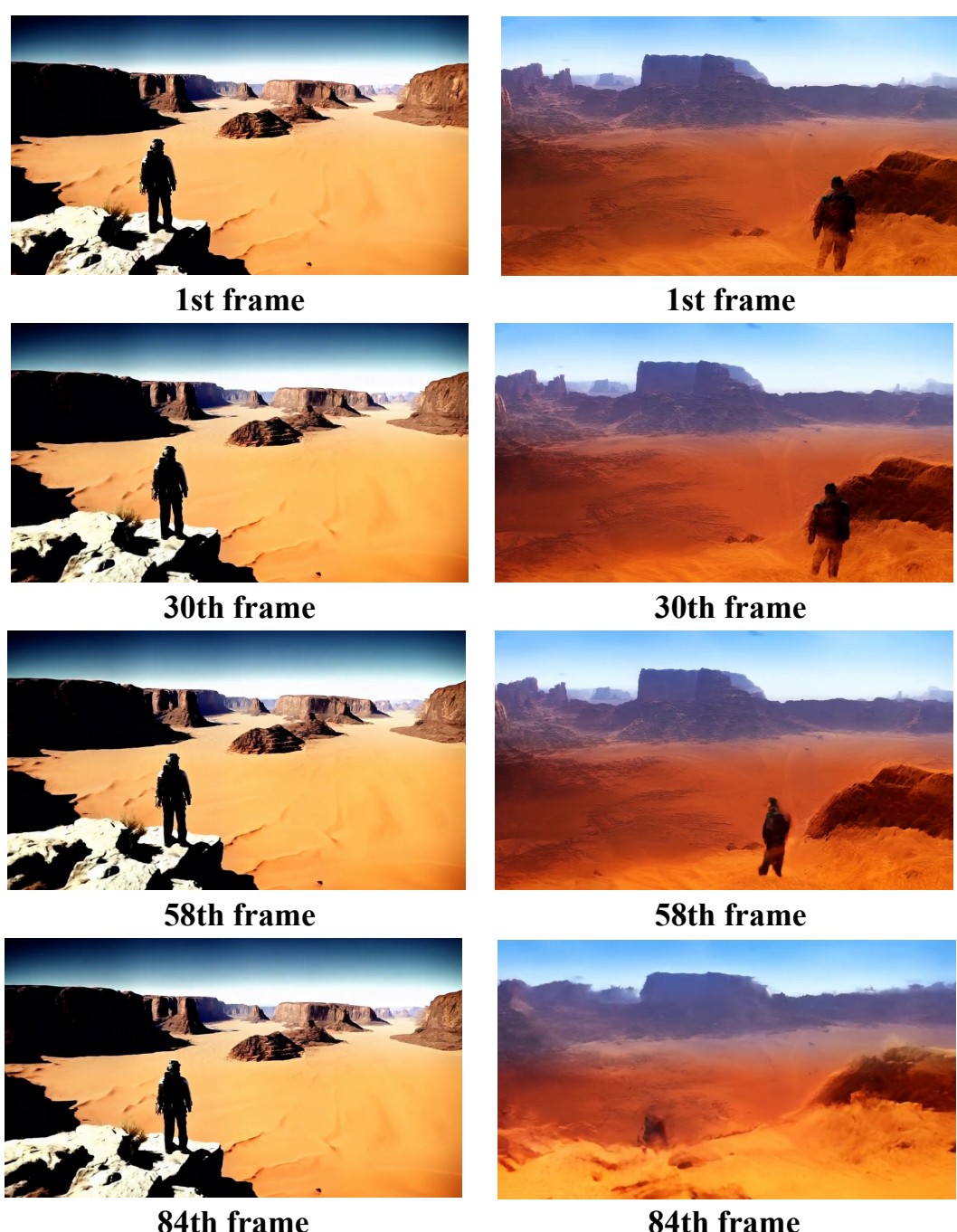

**1st frame**      **1st frame**

**30th frame**      **30th frame**

**58th frame**      **58th frame**

**84th frame**      **84th frame**

An explorer standing at the edge of a massive desert canyon, with swirling sands below and towering rock formations stretching into the distance.

Figure 13: Left: *IV-mixed Sampler*. Right: Vanilla Sampling.

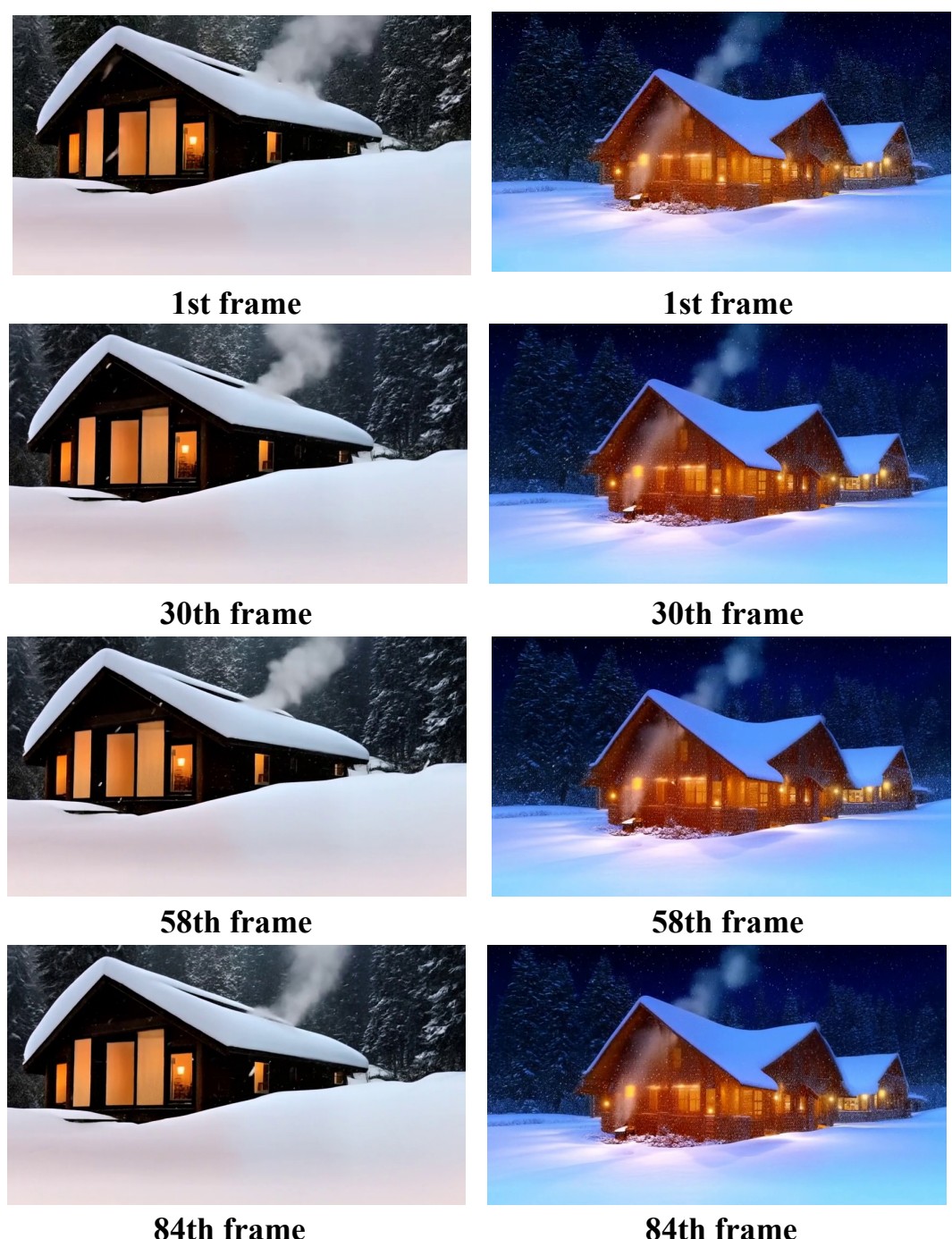

**1st frame** **1st frame**

**30th frame** **30th frame**

**58th frame** **58th frame**

**84th frame** **84th frame**

A cozy mountain cabin in the midst of a snowfall, with warm light emanating from the windows and smoke curling from the chimney.

Figure 14: Left: *IV-mixed Sampler*. Right: Vanilla Sampling.

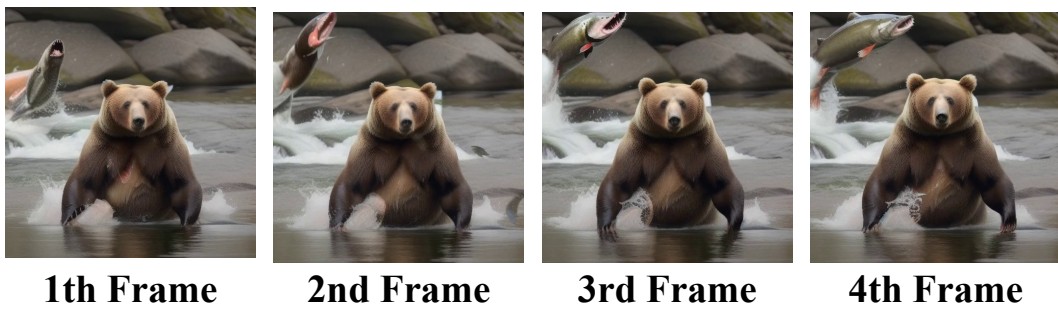

**1th Frame**  **2nd Frame**  **3rd Frame**  **4th Frame**

*a bear catching a salmon in its powerful jaws*

Figure 15: A selected example from VBench's subject consistency dimension.

## J VISUALIZATION

In order to avoid the size of the paper being too large for the reader, we downsample the video frames and present them here. We present the synthesized video visualization of Animatediff (SD V1.5, Motion Adapter V3) in Fig. 16-17, the synthesized video visualization of ModelScope-T2V in Fig. 18 and the synthesized video visualization of VideoCrafterV2 in Fig. 19.

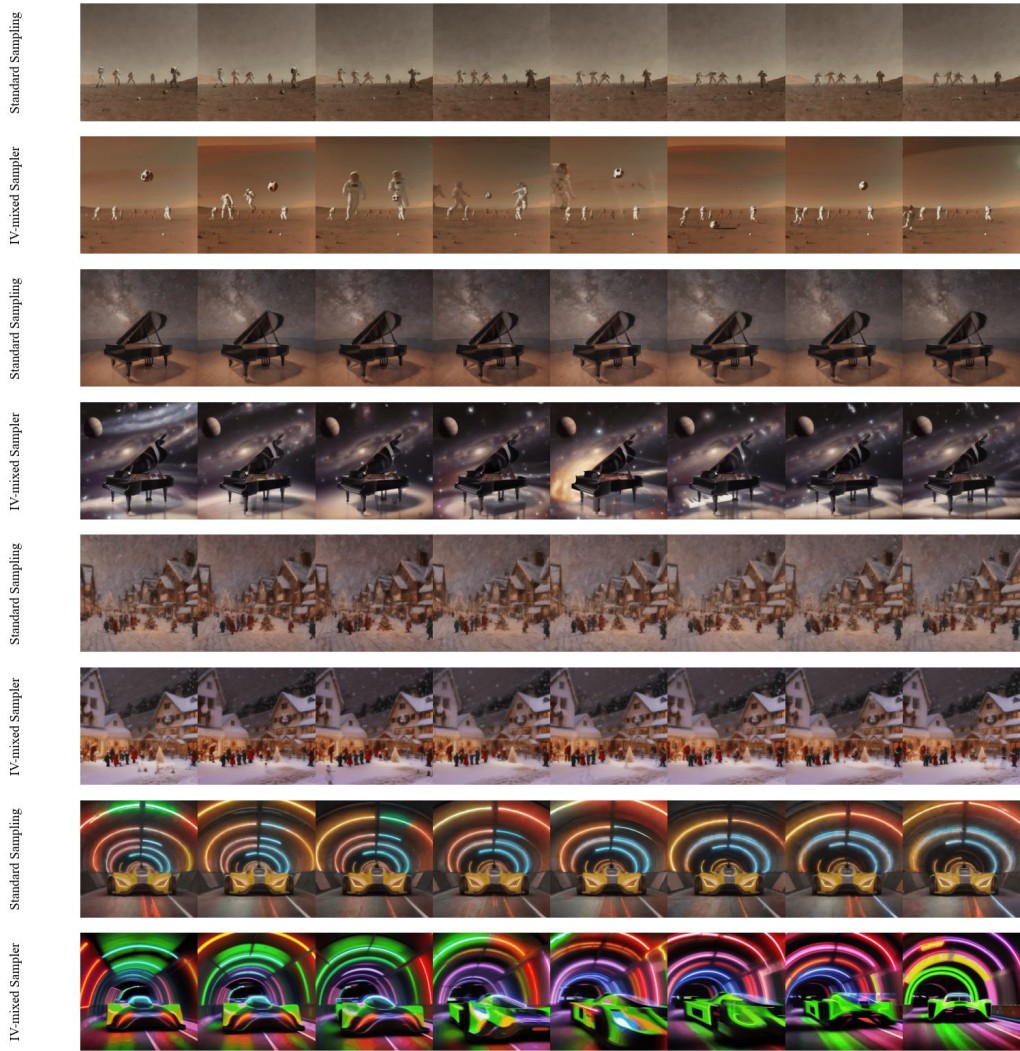

Figure 16: The synthesized video visualization of Animatediff (SD V1.5, Motion Adapter V3).

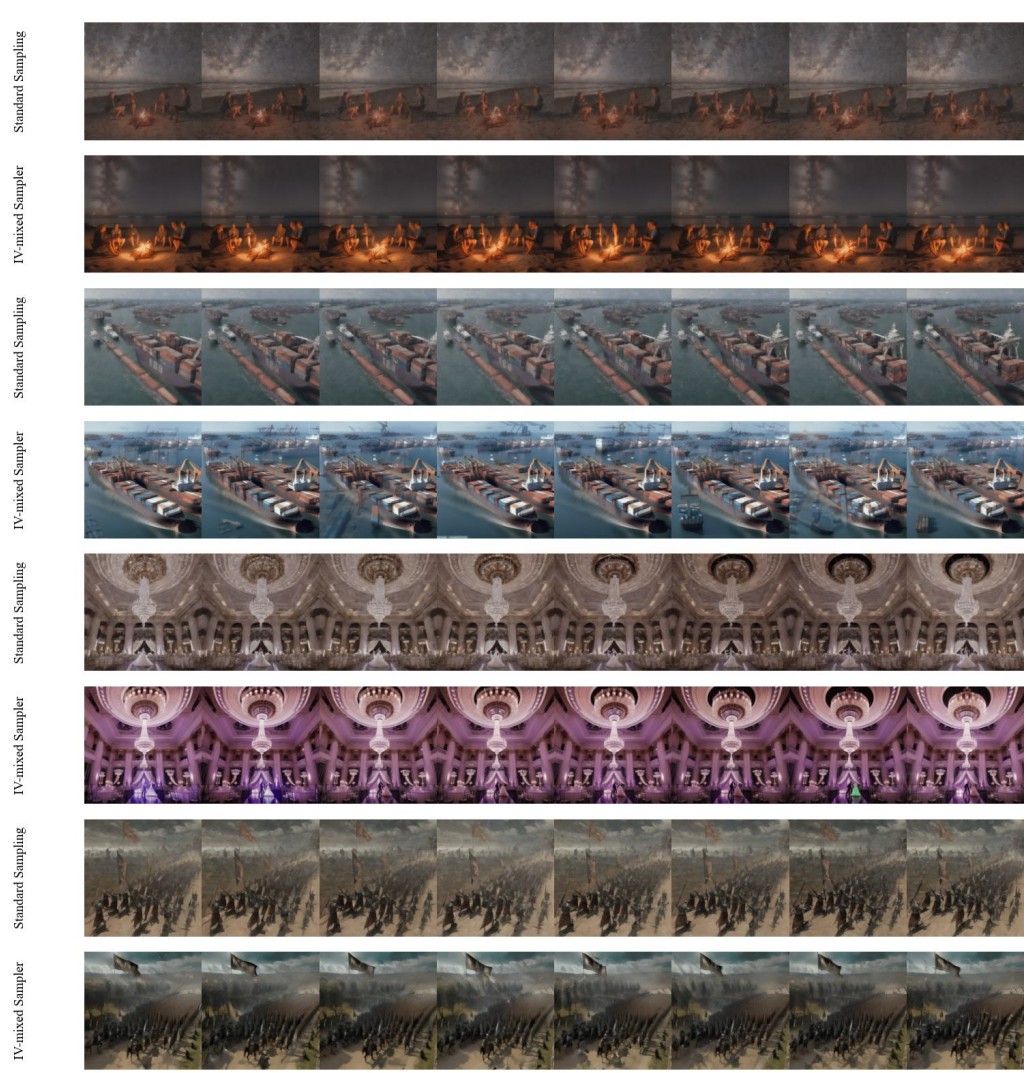

Figure 17: The synthesized video visualization of Animatediff (SD V1.5, Motion Adapter V3).

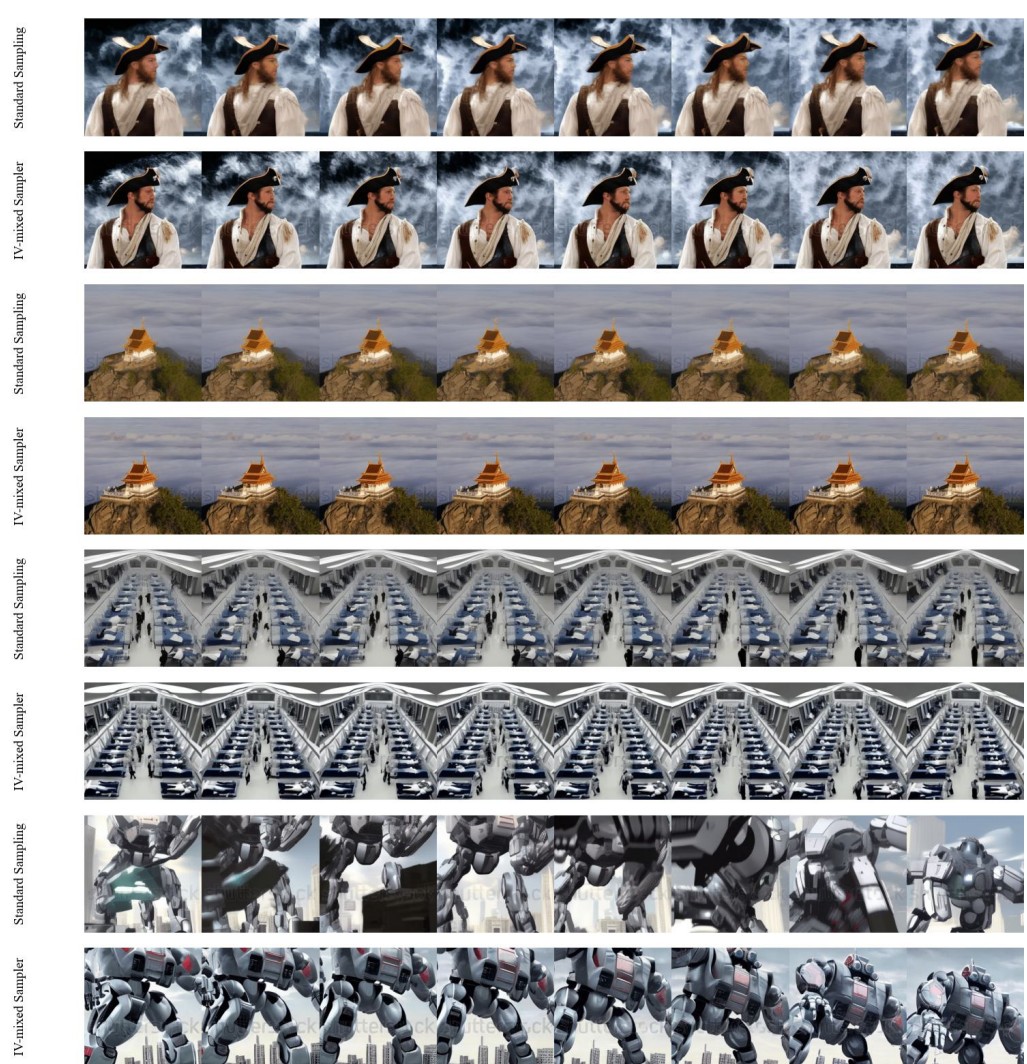

Figure 18: The synthesized video visualization of ModelScope-T2V.

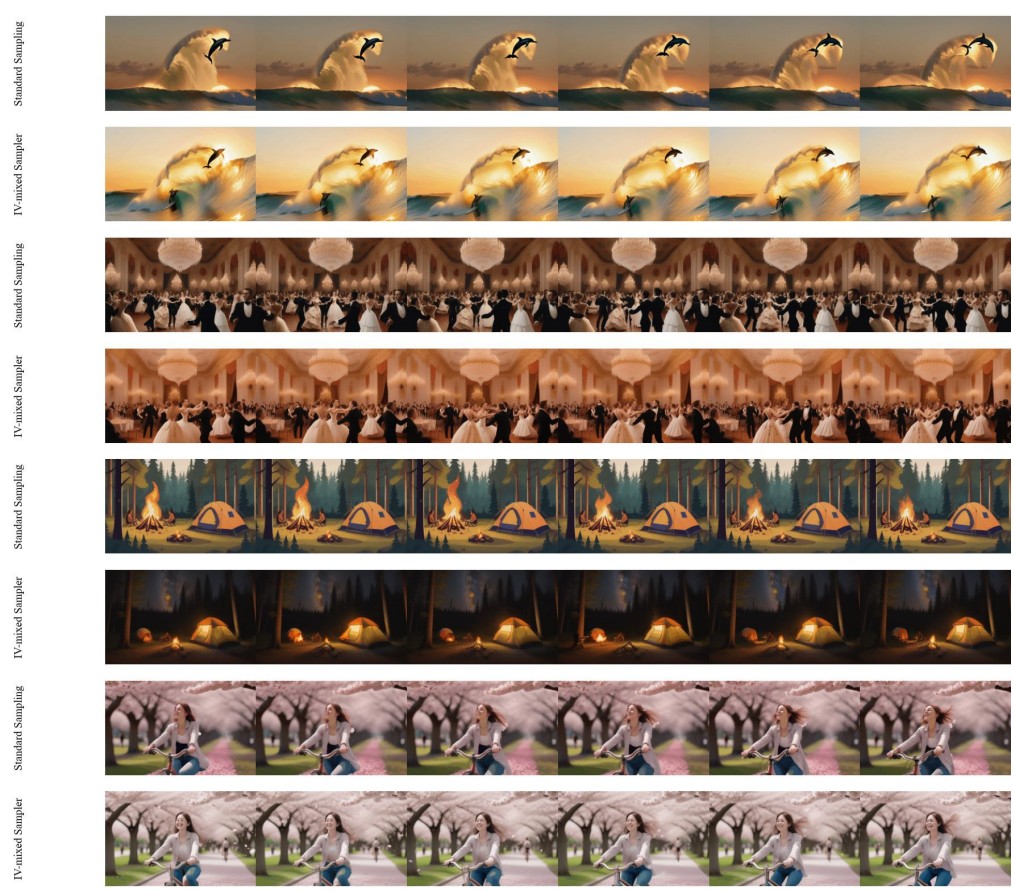

Figure 19: The synthesized video visualization of VideoCrafterV2.

