# OpenReview forum: "IV-mixed Sampler: Leveraging Image Diffusion Models for Enhanced Video Synthesis"
_ICLR.cc/2025/Conference — ICLR 2025 Poster_

### Official Review · Reviewer_ypZs · 2024-11-02

**Soundness:** 2
**Presentation:** 3
**Contribution:** 2
**Rating:** 6
**Confidence:** 3

**Summary:**

The paper introduces IV-Mixed Sampler, a novel algorithm designed to enhance video synthesis by leveraging the strengths of both Image Diffusion Models (IDMs) and Video Diffusion Models (VDMs). The core innovation is the use of IDMs to improve the quality of individual video frames while maintaining temporal coherence through VDMs during the sampling process. The authors claim state-of-the-art performance on four benchmarks, including UCF-101-FVD, MSR-VTT-FVD, Chronomagic-Bench-150, and Chronomagic-Bench-1649. The paper also provides a theoretical analysis of the IV-Mixed Sampler and its transformation into a standard inverse ODE process, as well as an exploration of the design space for hyperparameters.

**Strengths:**

Originality: The concept of combining IDMs and VDMs to enhance video synthesis is innovative. The paper addresses a significant gap in the field by improving the visual quality of synthesized videos while preserving temporal coherence.

Quality: The paper is well-structured, with a clear problem statement, methodology, experimental validation, and conclusion. The theoretical analysis provides a solid foundation for the proposed algorithm.

Clarity: The authors have done an excellent job of explaining complex concepts in a clear and concise manner. The figures and tables are well-designed and aid in understanding the content.

**Weaknesses:**

Evaluation Metrics: The paper employs benchmarks like UCF-FVD and MSR-VTT-FVD, which can not accurately assess text-to-video generation models. More suitable benchmarks like EvalCrafter or Vbench are needed for a comprehensive evaluation.

Demonstration Insufficiency: The provided demos do not clearly demonstrate the proposed method's superiority over baselines, requiring more compelling examples to showcase improvements.

Illustration Clarity: Figure 3's illustrations are not clear. The images presented do not effectively convey the differences in quality and temporal coherence, making it difficult for readers to grasp the paper's points regarding the performance of various samplers. High-quality, clear visualizations are crucial for helping readers understand the nuances of video synthesis methods, and the paper would benefit from improved clarity in its visual aids.

Limited Applicability: The method's compatibility with state-of-the-art video generation models, which often use distinct VAEs from image models, is limited. This restricts its potential applications and prospects in the field.

**Questions:**

1. What are the detailed reasons that we can't use different VAEs for IDM and VDM when adapting the proposed method? Do you have experiment results about this?
2. How does the IV-Mixed Sampler perform as the length and complexity of the video sequences increase? Are there any scalability issues that the authors have identified?
3. Potential Limitations: Are there any specific scenarios or types of video content where the IV-Mixed Sampler might underperform? If so, how might these limitations be addressed?

---

> ### Author Response · Authors · 2024-11-20
>
> 1. _**Vbench is needed for a comprehensive evaluation.**_
>
> | Animatediff-V2 | Average Score | subject consistency | temporal flickering | object class | multiple objects | human action |
> | :--- | :--- | :--- | :--- | :--- | :--- | :--- |
> | Vanilla Sampling | 60.19% | **95.30%** | **98.75%** | 90.90% | 36.88% | 92.60% |
> | **IV-mixed Sampler (ours)** | **66.69%** | 93.31% | 97.09% | **96.50%** | **58.46%** | **98.60%** |
> | **Animatediff-V2** | **color** | **spatial relationship** | **scene** | **appearance style** | **temporal style** | **overall consistency** |
> | Vanilla Sampling | 87.47% | 34.60% | 50.19% | 22.42% | 26.03% | 27.04% |
> | **IV-mixed Sampler (ours)** | **91.30%** | **59.78%** | **56.91%** | **24.44%** | **27.62%** | **29.54%** |
>
> > For a comprehensive evaluation of both visual quality and semantic consistency, we further assessed the performance of IV-mixed Sampler on VBench [1], with the results presented in the above table. From the above table, it is clear that IV-mixed Sampler outperforms vanilla sampling across most metrics, particularly on the multiple objects, where IV-mixed Sampler improved the performance of Animatediff from 36.88% to 58.46%. Additionally, the average scores of IV-mixed Sampler exceeded vanilla sampling by a margin of 6.5 points on Animatediff, fully demonstrating the effectiveness of IV-mixed Sampler.
>
>
> 2. _**Need to provide more compelling examples to showcase improvements**_
>
> > Thank you for pointing this out. We provide additional compelling examples of IV-mixed Sampler on different models on the anonymous website: [page](https://anonymous.4open.science/w/IVmixedSampler-A646/).
>
> 3. _**Figure 3's illustrations are not clear.**_
>
> > Thank you for your useful suggestion. We have redrawn Figure 3 (see the revised version) to use a more representative case, highlighting the low temporal coherence of IDM alone and the low quality of VDM alone.
>
> 4. _**The method's compatibility with state-of-the-art video generation models is limited. This restricts its potential applications and prospects in the field.**_
>
> > Thank you for your comment. Our proposed IV-mixed Sampler is compatible with the SOTA VDMs. In particular, IV-mixed Sampler, which relaxes the dependency on VAE, shows significant performance gains on the state-of-the-art VDM (12b parameters) Mochi-1 (see Appendix H in our revised version and the anonymous website [page](https://anonymous.4open.science/w/IVmixedSampler-A646/)). For example, for the prompt "A serene forest clearing at dawn, where deer graze peacefully while golden rays of sunlight pierce through the mist-laden trees", IV-mixed Sampler not only improves visual quality but also makes the deer in the video walk very naturally, which is really amazing!
>
> > Therefore, we believe that IV-mixed Sampler, now capable of improving video quality regardless of whether the IDM and VDM use the same or different VAEs, holds significant potential and promise in this field.
>
> 5. _**What are the detailed reasons that we can't use different VAEs for IDM and VDM when adapting the proposed method? Do you have experiment results about this?**_
>
> > This is an interesting question. In this revised version, we find it is possible to relax the dependency on VAE.  We can simply edit the video frames by transferring $x_t$ to $x_0$ in a small number of steps of DDIM and DDIM-Inversion. Details can be found in Appendix H.
>
>
> > Furthermore, for the IV-mixed Sampler presented in the original version, if the latent spaces of the VAEs in IDM and VDM are different, IV-mixed Sampler will not function properly.  We provide experimental results demonstrating this by applying SD XL as the IDM to Animatediff (SD V1.5, Motion Adapter V3), as shown in Fig. 10 of the revised version. As illustrated in Fig. 10 of the revised version, the resulting video frames are entirely black and cannot be recognized by the naked eye.
>
> 6. _**How does the IV-Mixed Sampler perform as the length and complexity of the video sequences increase? Are there any scalability issues that the authors have identified?**_
>
> >  IV-mixed Sampler continues to perform well. Specifically, our proposed IV-mixed Sampler shows significant performance gains on Mochi-1 for long video sequences (84 frames, fps=30). Note that the number of frames in Animatediff and Modelscope-T2V is 16. For further details, please refer to Appendix H in our revised version and the anonymous website  [page](https://anonymous.4open.science/w/IVmixedSampler-A646/).
>
>
> > We present the framework of IV-mixed Sampler on Mochi-1 in Figure 11 of the revised version, focusing on converting the video latent $x_t$ to $x_0$ using one-step sampling, enhancing it with the ControlNet Tile (SD XL), and adding a certain amount of random noise before passing it through Euler's inversion (similar to DDIM-Inversion). This approach allows the video latent to be converted back to $x_t$, thereby improving video fidelity through IDM.

---

> > ### Author Response · Authors · 2024-11-25
> > **Look forward to your post-rebuttal feedback!**
> >
> > Dear Reviewer ypZs,
> >
> > Thanks again for your insightful suggestions and comments. Since the deadline of discussion is approaching, we are happy to provide any additional clarification that you may need.
> >
> > In our previous response, we carefully addressed your feedback by:
> >
> > 1. Adding experiments on VBench.
> >
> > 2. Including more compelling examples on the anonymized page.
> >
> > 3. Relaxing the IV-mixed Sampler's dependency on the VAE, allowing it to be applied to the large-scale VDM Mochi-1.
> >
> > 4. Resolving the issue with the unclear Figure 3 and further explaining why the VAEs of IDM and VDM differ, which prevented the IV-mixed Sampler in the previous version of the paper from functioning correctly.
> >
> > We think these new experiments and explanations have clarified the contributions and strengthened our submission.
> > Please do not hesitate to reach out if there are any further clarifications or analyses we can provide.
> >
> > Thank you for your time and thoughtful feedback!
> >
> > Best, Authors

---

> > > ### Comment · Reviewer_ypZs · 2024-11-29
> > >
> > > Thank you, I appreciate your rebuttal, which addressed most of my concerns (except the results are still not that compelling, but it's a relatively novel idea and has some kind of effect anyway). I will raise the score to a weak accept.

---

> > > > ### Author Response · Authors · 2024-11-30
> > > >
> > > > Thank you for raising the score and for the valuable suggestion! We believe IV-mixed Sampler has been effective in improving the quality of the composite video. Of course, we will continue to explore better ways to leverage the IDM to enhance the performance of the VDM, and potentially customize an IDM specifically for the VDM in the future.

---

> ### Author Response · Authors · 2024-11-20
>
> 7. _**Potential Limitations: Are there any specific scenarios or types of video content where the IV-Mixed Sampler might underperform? If so, how might these limitations be addressed?**_
>
> > Thank you for your interesting question. IV-mixed Sampler is effective only if the quality of the images generated by IDM is higher than that of the video frames generated by VDM. Consequently, the algorithm may underperform if the VDM itself outperforms the IDM.  For example, in the case of Mochi-1-preview, if IDM SD V1.5 is not strong enough, it can simply be replaced with a stronger IDM, such as SD XL.
>
> [1] https://github.com/Vchitect/VBench.

---

### Official Review · Reviewer_ZanA · 2024-11-03

**Soundness:** 3
**Presentation:** 1
**Contribution:** 2
**Rating:** 5
**Confidence:** 3

**Summary:**

The paper proposes IV-Mixed Sampler, which is to combine image diffusion models (IDMs) with video diffusion models (VDMs) to get the benefit of higher image quality of IDM and good temporal consistency of VDM. This is achieved by sampling from x_t to x_{t+∆t} and from x_{t+∆t} to x_t using IDM and VDM

**Strengths:**

+ The paper combines theoretical proof with practical implementations, and is relatively well-rounded
+ The proposed method seems to make intuitive sense

**Weaknesses:**

The writing of the paper can be improved. Here are several comments but not just limited to these points. Overall the presentation is a bit confusing.
- Why do the paper keep mentioning OpenAI's strawberry in Abstract and Intro, which is closed-sourced and no paper or technical detais was released? How is this important or even related to motivating the paper?
- Figure 2 is confusing and should not be put in the intro. This is more like ablation studies, and it is made rather confusing, especially authors also start to talk about "R-" in the intro out of nowhere (line 94), and the starts to discussion "I-" and "V-" in line 107 without any explanation
- The use of "go", "back", "begin", "end" etc in the equations are also confusing
- Plots like Fig 6 are also hard to read

The results of the model is decent but not very strong. For example, FreeInit is significantly better than ours for UMT-FVD on ModelScope-T2V

**Questions:**

See weakness

---

> ### Author Response · Authors · 2024-11-20
>
> 1. _**Why do the paper keep mentioning OpenAI's strawberry in Abstract and Intro, which is closed-sourced and no paper or technical detais was released? How is this important or even related to motivating the paper?**_
>
> > Thank you for this question. We mention OpenAI's Strawberry because it exemplifies the inference scaling laws in large language models (LLMs). This motivates us to focus on improving the inference paradigm for video diffusion models, which could lead to significant performance gains. In our revised version, we would like to follow your suggestion to remove the mention of OpenAI's Strawberry from the abstraction and do not keep mentioning it. Instead, we focus on IDM's ability to enhance the visual quality of VDM and emphasize the inference-heavy aspects in the revised version.
>
> 2. _**Figure 2 is confusing and should not be put in the intro. There is no detailed explanation of “R-”, “I-” and “V-”.**_
>
> > We apologize for any distress caused by the figure. It has been moved to the ablation experiments section in the revised version. The meanings of “R-”, “I-” and “V-” are as follows:
>
> > + "R-": Forward noise addition using random noise addition.
> > + "I-": Forward noise addition using DDIM Inversion with IDM.
> > + "V-": Forward noise addition using DDIM Inversion with VDM.
>
> > We have added those explanation in the revised version.
>
> 3. _**The use of "go", "back", "begin", "end" etc in the equations are also confusing.**_
>
> > We apologize for any confusion caused by these definitions. Note that "go" and "back" denote the DDIM and DDIM-Inversion, respectively. We have renamed "begin" and "end" to t = 0 and t = 1 to more clearly indicate their meaning in the revised version. Specifically:
>
> > + $ \omega_{go}^{t=0} $represents the value of $ \omega_{go} $ at t = 0 (when DDIM is performed).
> > + $ \omega_{back}^{t=0} $ represents the value of $ \omega_{back} $ at t = 0 (when DDIM-Inversion is performed).
> > + $ \omega_{go}^{t=1} $ represents the value of $ \omega_{go} $ at t = 1 (when DDIM is performed).
> > + $ \omega_{back}^{t=1} $ represents the value of $ \omega_{back} $ at t = 1 (when DDIM-Inversion is performed).
>
>
> 4. _**Plots like Fig 6 are also hard to read.**_
>
> > Thank you for pointing this out. In the revised version, we have replaced Fig. 6 with three histograms and simplified some cases. Fig. 6 now more effectively illustrates the impact of IDM dynamic CFG scale, VDM dynamic CFG scale, and $\rho$ on the quality of the final composite video.
>
> 5. _**The results of the model is decent but not very strong. For example, FreeInit is significantly better than ours for UMT-FVD on ModelScope-T2V**_
>
> | Model | Method | Extra Model | UMT-FVD (↓) | UMTScore (↑) | GPT4o-MTScore (↑) |
> | --- | --- | --- | --- | --- | --- |
> | **ModelScope-T2V** | Standard DDIM | ❌ | 241.61$ _{\pm 0.002} $ | 2.65$ _{\pm 0.017} $ | 2.97$ _{\pm 0.013} $ |
> |  | FreeInit | ❌ | 220.96$ _{\pm 0.002} $ | **2.99**$ _{\pm 0.025} $ | **3.10**$ _{\pm 0.015} $ |
> |  | **IV-mixed Sampler (SD V1.5)** | ✅ | **234.90**$ _{\pm 0.001} $ (+6.71) | **3.00**$ _{\pm 0.022} $ (+0.35) | **3.16**$ _{\pm 0.035} $ (+0.19) |
> |  | **IV-mixed Sampler (RealVis-V6.0-B1)** | ✅ | **217.25**$ _{\pm 0.012} $ (+24.36) | **3.31**$ _{\pm 0.013} $ (+0.66) | **3.25**$ _{\pm 0.010} $(+0.28) |
>
> > IV-mixed Sampler outperforms both vanilla sampling and other comparison algorithms across nearly all models and metrics, indicating strong performance, and IV-mixed Sampler outperforms FreeInit on Modelscope-T2V using a stronger IDM. The performance of  IV-mixed Sampler largely depends on the generative capacity of the IDM and the VDM. In the previous version of this study, we used SD V1.5 as the IDM on Modelscope-T2V, but now, we choose RealVis-V6.0-B1 [1], which is less sensitive to resolution, as the IDM. We then adjust the CFG scales $\omega_\textrm{go}$ of the IDM and VDM to 2 ($\omega_\textrm{back}=-2$) and 3 ($\omega_\textrm{back}=-3$), respectively, and present the results in the above table. From these results, we observe that IV-mixed Sampler using RealVis-V6.0-B1 as the IDM outperforms FreeInit across all metrics, which highlights the potential of our algorithm.
>
> [1] https://huggingface.co/SG161222/Realistic_Vision_V6.0_B1_noVAE.

---

> > ### Author Response · Authors · 2024-11-25
> > **Look forward to your post-rebuttal feedback!**
> >
> > Dear Reviewer ZanA,
> >
> > Thanks again for your insightful suggestions and comments. Since the deadline of discussion is approaching, we are happy to provide any additional clarification that you may need.
> >
> > In our previous response, we carefully addressed your feedback by clarifying the diagrams, textual representations, and mathematical definitions. Additionally, we conducted experiments on Modelscope-T2V to demonstrate that Freeinit can be easily outperformed by IV-mixed Sampler with a better IDM.
> >
> > We think these new experiments and explanations have clarified the contributions and strengthened our submission.
> > Please do not hesitate to reach out if there are any further clarifications or analyses we can provide.
> >
> > Thank you for your time and thoughtful feedback!
> >
> > Best, Authors

---

### Official Review · Reviewer_ViGW · 2024-11-05

**Soundness:** 3
**Presentation:** 3
**Contribution:** 3
**Rating:** 8
**Confidence:** 4

**Summary:**

The paper presents a novel sampling scheme for video diffusion models that leverages pre-train image diffusion models, as they generally have higher visual fidelity compared to current open source video models. The sampling method takes steps forward / backward (DDIM inversion / DDIM) in the diffusion model, alternating between the score functions of the image and video models. Results show high fidelity visual quality while retaining consistent motion in generated video.

**Strengths:**

* The paper is generally clear, and well written
* The proposed method is novel and interesting
* Results seem to show good improvement in generation quality of the videos, while retaining consistent motion
* Main experiments and ablations are thorough and show the benefits and tradeoffs of different instantiations of the proposed method

**Weaknesses:**

* The sampling process requires both I/V models to be in the same underlying latent space, which may be restrictive. How would this method also be used in cases where there is temporal downsampling in the video latent space (as this this is a very common video generation architecture)?
* It is unclear how useful / relevant this method may be ~6+ months from now, as the main motivation of the paper is leveraging the lack of good open source video models, and public video datasets will get better (e.g. OpenVid10M [1]), and better video generation models will be released (e.g. Mochi-1 [2] as of recently).

[1] https://huggingface.co/datasets/nkp37/OpenVid-1M
[2] https://huggingface.co/genmo/mochi-1-preview

**Questions:**

* What is the variance of the quantitative metrics (e.g. FVD in Tables 1 and 2)? The values are pretty close and it’s unclear  if the results are statistically significant
* It would be nice to have more video samples to look at (e.g. on an anonymous website). There is only one set of videos in the supplementary, and it is hard to see motion for ones in the paper appendix.
* Does this method still work in scenarios using distilled models? E.g. 1 step or 2 step generation

---

> ### Author Response · Authors · 2024-11-20
>
> 1. _**How would this method also be used in cases where there is temporal downsampling in the video latent space**_
>
> > Thank you for your interesting question. In our revised version, we relaxed the dependency of IV-mixed Sampler on VAE, allowing it to operate on different VAEs for IDMs and VDMs (see the performance on Mochi-1 in the anonymous website [page](https://anonymous.4open.science/w/IVmixedSampler-A646).
>
>
> > Furthermore, we can apply IV-mixed sampler by treating the VDM itself as an IDM. To improve the visual quality of the synthesized samples from the IDM (i.e., the VDM), high-quality images (with data of $ b \times c \times 1 \times h \times w $) can be used by the DPO to fine-tune the VDM.
>
>
>
> 2. _**The method's relevance may decline in 6+ months as public video datasets (e.g., OpenVid10M) and video generation models (e.g., Mochi-1) improve.**_
>
> > The potential of IV-mixed Sampler will remain strong beyond 6+ months. The concerns you raise are very practical. However, even the 12b open-source large-scale VDM Mochi-1 still does not match the visual quality (the video suffers from severe distortion) of middle-scale IDM (such as SD XL). Our proposed IV-mixed Sampler, which we have slightly modified to eliminate dependency on VAE, shows significant performance gains on Mochi-1 (see Appendix H in our revised version and the anonymous website [page](https://anonymous.4open.science/w/IVmixedSampler-A646/). For example, for the prompt "A serene forest clearing at dawn, where deer graze peacefully while golden rays of sunlight pierce through the mist-laden trees", IV-mixed Sampler not only improves visual quality but also makes the deer in the video walk very naturally, which is really amazing!
>
>
>
> > The changes made to Mochi-1 are shown in Figure 11 of the revised version, focusing on converting the video latent $x_t$ to $x_0$ using one-step sampling, enhancing it with the ControlNet tile (SD XL) [3], and adding a certain amount of random noise before passing it through Euler's inversion (similar to DDIM-Inversion). This approach allows the video latent to be converted back to $x_t$, thereby improving video fidelity through IDM.
>
>
> > Therefore, we believe that IV-mixed Sampler, which is now capable of improving video quality whether the IDM and VDM use the same VAE or different VAEs, still holds great potential for future development.
>
>
>
> 3. _**What is the variance of the quantitative metrics (e.g. FVD in Tables 1 and 2)?**_
>
> > Thank you for your meaningful suggestion. We have included the standard deviations for Tables 1, 2, 3, and 4 in the revised version. Specifically, we sampled and evaluated each configuration three times (set seeds 41, 42, and 43), and found that the standard deviations of all results are on the order of two decimal places. This suggests that the improvement from the IV-mixed sampler is significent.
>
>
> 4. _**It would be nice to have more video samples to look at (e.g. on an anonymous website).**_
>
> > Thank you for your valuable advice. We have presented the results of IV-mixed Sampler on different models on the anonymous website: [page](https://anonymous.4open.science/w/IVmixedSampler-A646/).

---

> ### Author Response · Authors · 2024-11-20
>
> 5. _**Does this method still work in scenarios using distilled models? E.g. 1 step or 2 step generation.**_
>
> | _**Model**_ | _**Method**_ | _**Extra Model**_ | _**UMT-FVD (↓)**_ | _**UMTScore (↑)**_ | _**GPT4o-MTScore (↑)**_ |
> | --- | --- | --- | --- | --- | --- |
> | **MCM**   **(Animatediff V2)** | _**Standard LCM**_ | ❌ | _**277.91**_$ _{\pm 0.025} $ | _**1.98**_$ _{\pm 0.012} $ | _**2.66**_$ _{\pm 0.021} $ |
> |  | **IV-mixed Sampler**_**(ours)**_ | ✅ | _**246.59**_ $ _{\pm 0.021} $   **(+31.32)**  | _**2.19**_ $ _{\pm 0.013} $   **(+0.21)** | _**2.87**_$ _{\pm 0.009} $   **(+0.21)**|
>
>
> > Yes, IV-mixed Sampler still work using distilled models. To verify the effectiveness of IV-mixed Sampler on the distillation-based accelerated sampling model, we apply IV-mixed Sampler to the motion consistency model (MCM) [2]. We use the Animatediff-laion version from the official MCM library and set the number of sampling steps to 4. After extensive empirical experimentation, we found that the sampling mechanism shown in Fig. 9 of the main paper can enhance MCM performance by introducing IDM. Specifically, we used three checkpoints: the VDM w/o distillation, the VDM w/ distillation, and the standard IDM. During each sampling step with the VDM w/ distillation, IV-mixed sampling is performed using the VDM w/o distillation and the standard IDM. It is important to note that if the VDM w/ distillation is used for IV-mixed sampling (i.e., replacing the VDM w/o distillation with the VDM w/ distillation), it will not work due to its LCMScheduler at each step, which is non-deterministic and would disrupt the semantic information provided by the IDM.
>
>
> > We present the results in the above table, and we can find that IV-mixed Sampler significantly improves the performance of MCM in all metrics.
>
>
>
> [1] I4VGen:ImageasStepping StoneforText-to-Video Generation, Arxiv 2024.
>
> [2] Motion Consistency Model: Accelerating Video Diffusion with Disentangled Motion-Appearance Distillation, NeurIPS 2024.
>
> [3] https://huggingface.co/OzzyGT/SDXL_Controlnet_Tile_Realistic.

---

> > ### Comment · Reviewer_ViGW · 2024-11-21
> > **Response**
> >
> > Thank you for the detailed feedback and experiments. I think most of my concerns have been addressed (main ones were applicability to other VAEs, and for distilled models), so I will raise my score to an Accept.
> >
> > A few other notes:
> > - The website shows examples, but the videos don't always play for me, specifically subsets of the videos with Before / After dragging interface (e.g. for "A group of friends sitting around a campfire..." or "Two horses racing..."). I am using Chrome
> > - It may also be good to eventually show some failure models of the method - from the metrics it is unclear if it's always slightly better, or for some examples a lot better and other examples worse (with it being slightly better on average). From the VBench results it may be that the method causes a little bit of worse temporal consistency in exchange for high visual fidelity, but would be good to provide more details / insight on that front.

---

> > > ### Author Response · Authors · 2024-11-22
> > >
> > > Thank you for your kind words, thoughtful comments, and for raising the score! We also appreciate your further constructive feedback. Here is our response:
> > >
> > > 1. Regarding the issue of some videos not displaying on the anonymous page, it may be due to the time required for the video to load. We were able to display it correctly using Chrome, though it required a long wait.
> > >
> > > 2. We find that slight inconsistencies can arise when the magnitude of motion change is too large across a few consecutive frames. This situation is discussed further in Appendix I (our latest revised version).

---

### Author Response · Authors · 2024-11-20

We appreciate all reviewers for their positive comments:

1. The paper is generally clear, and well written. This paper is also well-structured, with a clear problem statement, methodology, experimental validation, and conclusion **[ViGW, ypZs]**.

2. The method  is novel and interesting, which seems to make intuitive sense **[ViGW, ZanA]**.

3. The paper combines theoretical proof with practical implementations, and is relatively well-rounded **[ZanA]**.

4. Main experiments and ablations are thorough and show the benefits and tradeoffs of different instantiations of the proposed method **[ViGW]**.

We also make the revision following your constructive suggestions:

1.  Exploring the potential of IV-mixed Sampler on more advanced VDMs (e.g., Mochi-1) and addressing scalability issues for longer videos **[ViGW, ypZs]**.

2. Resolving presentation issues throughout the paper, including figures, explanations, and definitions of formulas **[ZanA]**, evaluating the IV-mixed Sampler on VBench **[ypZs]**.

3. Assessing the performance of the IV-mixed Sampler on the distilled model **[ViGW]**.

4. Further validation on VBench **[ypZs]**.

5. Adding an anonymous page to present a visualization of IV-mixed Sampler **[ViGW]**.

6. Add the standard deviation to indicate IV-mix sampler is significant **[ViGW]**.

7. Replace IDM to allow IV-mixed Sampler to beat FreeInit on Modelscope-T2V **[ZanA]**.

These suggestions will certainly help us improve the quality of the paper, and we will incorporate all of them in our revision. Below, we summarize our responses and provide further clarifications to the questions from each reviewer.

---

### Meta-Review · Area_Chair_jfQ8 · 2024-12-24

**Metareview:**

The paper proposes IV-mixed Sampler, a method for improving the quality and capabilities of video diffusion models by integrating an image diffusion model in the sampling process (since image models are known to have higher visual quality)---effectively mixing the spatial/visual fidelity of an image model with the temporal coherence of a video model. Strengths include the practical value of improving video synthesis without requiring training and comprehensive experimental validation showing improvements over existing methods (state-of-the-art performance across multiple benchmarks). Reviewers questioned whether the improvements were significant enough compared to existing approaches, and raised concerns about limited applicability (since the method requires two models that have been trained on the same latent space). Additional weaknesses include insufficient real-world testing and unclear explanations of some technical components.

Reviews were mixed, but tend positive. Given the reasonable method and compelling results, I am inclined to recommend acceptance.

**Additional Comments On Reviewer Discussion:**

Reviewers raised concerns about the method's compatibility with different latent spaces and the need for more comprehensive evaluation metrics. The authors responded by demonstrating compatibility with newer models, providing additional evaluation results, and showing stronger performance when using better image models. Reviewers were satisfied with these clarifications, and increased their scores to positive. One reviewer did not respond, and their review takes less weight in this final recommendation.

---

### Decision · Program_Chairs · 2025-01-22

Accept (Poster)